# Optimizing Retrieval for RAG
# via Reinforcement Learning

**Jiawei Zhou    Lei Chen**
The Hong Kong University of Science and Technology
The Hong Kong University of Science and Technology (Guangzhou)
`{jzhoubu,leichen}@connect.ust.hk`

## Abstract

As retrieval-augmented generation (RAG) becomes more widespread, the role of retrieval is shifting from retrieving information for human browsing to retrieving context for AI reasoning. This shift creates more complex search environments, where relevance is difficult to pre-define. Existing retrievers rely on supervised fine-tuning (SFT) with human labels or synthetic data, resulting in static relevance that struggles to adapt to diverse RAG environments. To address this challenge, we propose **R3**, a **R**etrieval framework optimized for **R**AG through **R**einforcement learning (RL). Specifically, we adopt an RL training paradigm that enables the retriever to explore and self-improve within given RAG environments, automating the learning process with minimal manual experimentation or tuning effort. Extensive experiments across diverse tasks demonstrate that R3 improves RAG performance by 5.2% over the original retriever and surpasses state-of-the-art retrievers by 4.9%, while achieving comparable results to LLM-augmented retrieval and RAG systems built on post-trained or instruction-tuned LLMs. It is both efficient and practical, requiring only 4 GPUs and completing training within a single day.

## 1 Introduction

Retrieval-augmented generation (RAG) [1, 2, 3] is a well-established paradigm that integrates large language models (LLMs) [4] with information retrieval (IR) [5] to access external knowledge. Recently, RAG has rapidly evolved into a foundation for a new wave of advanced AI applications [6]. Examples include web-browsing agents [7, 8] that interact with live web content, conversational chatbots [9, 10, 11] that integrate retrieval to enable knowledge-grounded and emotionally aware dialogue, and DeepResearch assistants [12, 13, 14] that autonomously analyze and synthesize information across multiple sources to produce comprehensive reports.

These RAG applications signify a fundamental shift in the role of retrieval, from search for humans to search for AI. As illustrated in Figure 1, traditional **IR** searches information for human users, where relevance is pre-defined as superficial similarity to ensure interpretability [15, 16], so that human users can instantly grasp the underlying relations. In contrast, **RAG** searches contextual knowledge for AI systems, which can efficiently process and reason over vast amounts of information and are designed to handle diverse and evolving tasks. The retriever functions as an internal component within a RAG environment that encompasses tasks, workflows, LLMs, and other components, making relevance more environment-dependent, hard to pre-define, and necessary to explore.

Existing RAG applications have several retrieval options. Some leverage search engine APIs, which are convenient but expensive, difficult to scale, and cannot be further optimized [17]. Others deploy neural retrievers locally. However, directly applying these off-the-shelf retrievers can suffer from **gap between IR and RAG**. Studies have shown that traditional IR often retrieve semantically related but unhelpful content [18, 19, 20], LLM-generated documents [21], or misleading evidence [22],

39th Conference on Neural Information Processing Systems (NeurIPS 2025).

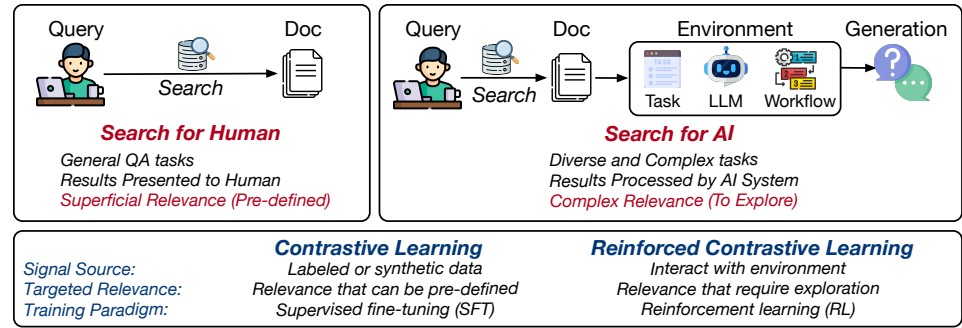

Figure 1: Comparison of traditional IR setting and RAG setting.

all of which can degrade RAG performance. Furthermore, Cuconasu et al. [23, 24] demonstrate that while traditional IR retrieves documents containing the correct answer, these documents do not necessarily improve RAG performance. Additionally, related findings have emerged in the study of LLM faithfulness, where Anthropic [25] and Apple [26] found that chain-of-thought prompts often fail to reflect the actual reasoning process of LLMs, and even advanced LLMs sometimes fail to follow explicit instructions. These observations reveal a key challenge of context engineering: there is no clear guidance on what constitutes effective context across different environments, leaving the process heavily dependent on manual effort.

In addition to the studies mentioned above, we further examine this gap through empirical experiments presented in Appendix D, where we evaluate the varying capabilities of different retrievers (unsupervised, supervised, and SOTA) on tasks of varying complexity in both IR and RAG settings. In IR, we examine whether the retriever can retrieve documents containing the answer, while in RAG, we assess whether the retrieved document can prompt the LLM to generate the correct answer. The results reveal two key findings. **Finding 1: Better IR performance does not always lead to better RAG performance, especially when a task shift occurs.** For QA tasks, we observe that higher accuracy in IR settings generally correlates with better RAG performance. However, this trend does not hold for non-QA tasks. For example, on PubHealth, a vanilla DPR$_{MS}$ outperforms a SOTA retriever in RAG settings, despite being weaker in IR metrics. Similarly, on ARC, the unsupervised CONTRIEVER achieves the best RAG performance, outperforming its supervised counterparts. **Finding 2: Substantial room for improvement in retrieval within the RAG system.** We find that although most required knowledge already exists in the datastore, each retriever succeeds on only a subset of queries. For example, on NQ, 77% of queries can be answered by at least one retriever via RAG, yet the best single retriever covers only 43%.

To bridge the gap from IR to RAG, we propose **R3**, a **R**etrieval framework optimized for **R**AG via **R**einforcement learning. Unlike prior works that **learn pre-defined relevance via supervised fine-tuning (SFT)** through annotated or synthetic data, R3 aims to **explore relevance via reinforcement learning (RL)** within the RAG environment. Specifically, R3 follows the standard RAG workflow: it retrieves documents and incorporates them as context for generation. Based on the generated outcomes, the retrieved documents are labeled as positive or negative with respect to the query, and contrastive learning objective is applied to optimize the retriever accordingly. We refer to this paradigm as reinforced contrastive learning, where contrastive signals arise from interaction with environment rather than manual annotation. Extensive experiments demonstrate that R3 enables retrievers to effectively adapt from IR to RAG, achieving an average 5.2% improvement in 1-shot accuracy and consistently outperforming SOTA retrievers as well as LLM-empowered retrieval methods.

Our contributions are summarized as follows:

1. We identify and analyze the relevance gap between traditional IR and RAG settings, supported by evidence from prior studies across diverse domains as well as our own empirical findings.

2. We propose R3, which pioneers reinforcement learning optimization for embedding-based retrievers in RAG systems.

3. R3 is simple and effective, delivering consistent RAG improvements with modest training costs, consistently outperforming existing SOTA retrievers and achieving improvements comparable to those seen in post-trained or instruction-tuned LLM across several tasks.

## 2 Related Works

**The Evolution of Information Retrieval.** Information retrieval has long been a core mechanism for various applications. The field began with heuristic approaches such as TF-IDF and BM25 [27], which have been widely adopted in both research and industry. As the field progressed, the advent of neural networks gave rise to neural retrieval [28]. Yet, these models still heavily relied on training data annotated using BM25 and manual validation [29, 30], leading to superficial relevance. In recent years, however, the rise of RAG has exposed the gap between search for humans and search for AI. This shift has prompted efforts to optimize retrieval models specifically for RAG. Early work in this direction focus on learning-free methods, such as using LLMs or heuristics to determine when to retrieve [31, 32], performing adaptive retrieval with reasoning [33, 34, 35], or decomposing and refining complex queries [36, 37]. Other research explored learning-based approaches, where retrievers are optimized jointly [2, 38] or separately [39, 40] with LLMs, using unsupervised objective like prefix language modeling [40], or supervised objective that maximize question-answer generation likelihood [41]. Despite their successes, these methods often come with high computational costs and focus primarily on general relevance, with limited consideration of specific RAG environments.

Beyond conventional RAG systems that search context for LLMs, there are many more complex, real-world environments where retrieval must interact with other intelligent components to address emerging challenges. For example, searching multi-modal data to improve multi-modal LLM reasoning [42, 43], retrieving MCP services for agents [44], or searching action modules or planning for robot embodiment [45, 46]. All of these emerging and advanced applications highlight complex environment-specific relevance. Our method, in line with this area, provides a solution by efficiently training specialized retrievers tailored to a specific environment for applications.

**Contrastive Learning.** Contrastive learning (CL) [47, 48] is a well-established method used to train bi-encoder retrievers by pulling representations of positive pairs closer and pushing negatives apart [28, 49]. CL relies on contrastive labels, which define the positive or negative relationship between pair-wise data. These relational data (i.e., positive and negative query–document pairs) can be obtained through annotation [29], extracted from the web [50, 51], or, more recently, synthesized using LLMs [52, 53]. While these contrastive labels provide effective training signals, they are inherently defined at the data level, making them difficult to transfer across different environments when needed. If components such as tasks, LLMs, or workflows within the environment change, these labels often lose effectiveness and require reconstructed. We further introduce **Reinforced Contrastive Learning (RCL)**, which distinguishes itself from conventional CL by constructing contrastive signals on-the-fly in a trial-and-feedback manner. Simply put, traditional CL involves SFT on annotated data to learn pre-defined relevance, while RCL leverages RL to explore proper relevance that suit the specific search environment. Our framework follows RCL paradigm, enabling versatility and automatic generalization to any RAG environment.

## 3 Methodology

In this section, we introduce **R3**, a framework that optimizes **R**etrieval for **R**AG via **R**einforced constrastive learning. While our idea is straightforward, it presents several challenges. In the following sections, we detail the problem setup (§3.1), the solution to the first challenge, which addresses the on-policy retrieval issue (§3.2), the solution to the second challenge, which focuses on mitigating the cost of autoregressive generation (§3.3), and the reinforced contrastive learning (§3.4).

### 3.1 Problem Setup

A $RAG$ framework typically consists of:

- Retriever $\mathcal{R}_\theta$ parameterized by $\theta$;
- Datastore $\mathcal{D}$ containing a vast number of documents $d$;
- User query $q$, and the corresponding answers $a$, if provided;
- RAG environment $env$, which refers to everything surrounding the $\mathcal{R}_\theta$, such as LLM $\mathcal{G}$, task instructions, and specific workflows, etc.;
- Reward function $Reward(\cdot)$ that evaluates the quality of the generated response.

The downstream RAG pipeline generally follows:

1. **Retrieval**: retrieve the top-$k$ relevant documents from the $\mathcal{D}$, with a relevance function $f_\theta$:

$$\{\hat{d}\}_k = \mathcal{R}_\theta(q, \mathcal{D}, k) \triangleq \underset{d \in \mathcal{D}}{\mathrm{argmax}_k} f_\theta(q, d)$$

2. **Generation**: generate response based on the retrieval results and the environment $env$.

$$\hat{y} = \mathcal{G}_{env}(q, \{\hat{d}\}_k) \triangleq RAG_\theta(q \mid env)$$

3. **Evaluation**: measure the quality of the generation $\hat{y}$. In our setup, we measure whether the generation contains the answer, which can be considered as a binary reward:

$$Reward(\hat{y}) = \begin{cases} 1 & \text{if } \hat{y} \text{ contain } a, \\ 0 & \text{otherwise.} \end{cases}$$

**Optimization Goal.** Given a RAG environment, R3 aims to learn the retrieval parameters to maximize the system reward for all training queries:

$$\begin{aligned} \hat{\theta} &= \underset{\theta}{\mathrm{argmax}} \sum_{\forall q} Reward(\hat{y}) \\ &= \underset{\theta}{\mathrm{argmax}} \sum_{\forall q} Reward(RAG_\theta(q \mid env)) \end{aligned}$$

**Overview.** Our work applies reinforcement learning (RL) to learn embedding-based retrievers, where downstream rewards are used to ensure consistent improvements. A key challenge lies in the interaction cost of retrieval within the environment, given that retrieval is only one small step in the overall RAG pipeline. The main contribution of this paper is to make this interaction cost manageable, using various approximations and designed retrieval architectures. We position our work as leveraging these techniques to automate retrieval learning for RAG with stable improvement, aiming to achieve a better trade-off between cost and effectiveness in practical applications.

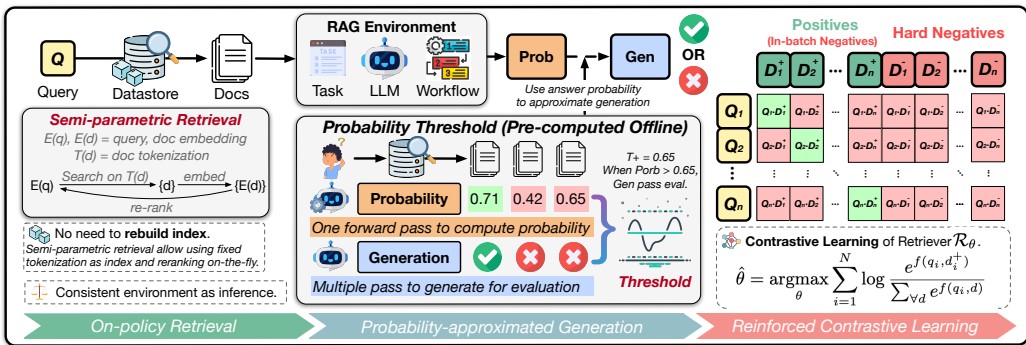

Figure 2: Illustration of the R3 training process.

**Training Pipeline.** Our training consists of three steps: (i) on-policy retrieval that conduct search during training with its latest parameters; (ii) approximiated LLM generation based on the retrieved documents; (iii) reinforced contrastive learning to optimize the retriever. Steps i and ii present practical challenges, which we address in the next section. Both are essential to enable effective reinforced contrastive learning. An illustration of our framework is depicted in Figure 2.

### 3.2 On-Policy Retrieval

In this step, given a query $q$, we use the on-policy retriever to retrieve the top-$k$ relevant documents. The main challenge in this stage concerns the retrieval index.

**Challenge of Index Staleness.** As is well known, a bi-encoder retriever consists of a query encoder $E_\theta^Q$ and a document encoder $E_\theta^D$. During training, as the parameters update from $\theta \to \theta'$, query

embeddings can be computed on the fly using the updated encoder $E_{\theta'}^Q$, since the number of queries is limited to the batch size. In contrast, the document datastore typically contains billions or even trillions of documents (i.e., $|\mathcal{D}| \gg 1$), making it impractical to recompute document embeddings with the updated encoder $E_{\theta'}^D$. As a result, the index $E_\theta^D(D)$ becomes stale as training progresses $\theta \to \theta'$. Fundamentally, this challenge stems from the coupling between index and parameters $\theta$.

**Prior solutions** alleviate index staleness by periodically rebuilding the index [2, 40] or freezing the document encoder during training [41, 38]. While shorter re-index intervals are observed to yield better results [54], longer intervals are often chosen to reduce re-indexing costs. Although these strategies alleviate index staleness to some extent, residual staleness remains and ultimately limits the effectiveness of retriever training.

**Our Solution.** We adopt our recently proposed semi-parametric retriever architecture SɪDR [55], which is designed to address this issue.

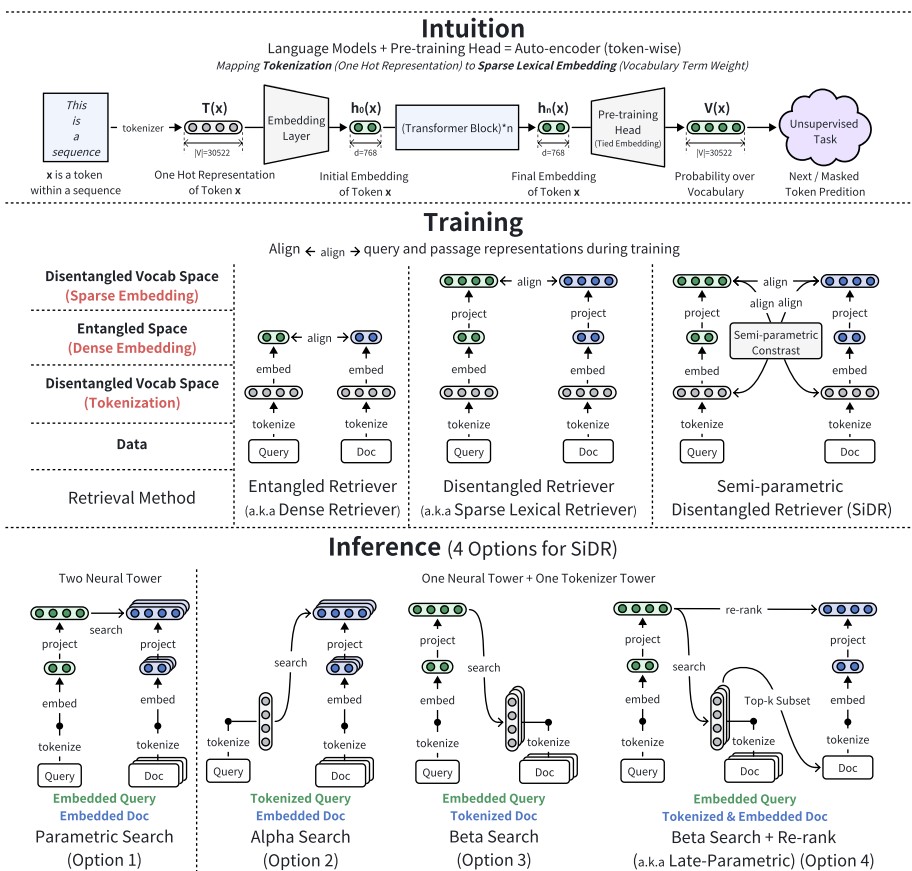

Figure 3: Overview illustration of SɪDR.

**Revisiting SɪDR.** Figure 3 illustrates the core idea of SɪDR. Intuitively, a language model together with its pre-training head can be viewed as an auto-encoder that encodes one-hot token representations $T(x) \in \{0, 1\}^{|V|}$ into softmax vocabulary probabilities $V(x) \in (0, 1)^{|V|}$ to predict masked or next tokens. Inspired by this, SɪDR aligns bag-of-token representation with sparse embeddings so that, at downstream, the embedded query can directly search over tokenized documents, thereby decoupling the index from parameters $\theta$.

Specifically, SɪDR builds upon a BERT-based models with three simple modifications to its pre-training head: (i) replacing the softmax with elu1p activation; (ii) aggregating token representations via max pooling; and (iii) applying top-$k$ sparsification. Given a sequence $X$, SɪDR embeds it into a $|V|$-dimensional sparse embedding which we refer to $E_\theta(X) \in (0, +\infty)^{|V|}$. During training, SɪDR applies constrastive learning across three channels: $E_\theta(q)$ with $E_\theta(d)$, $T(q)$ with $E_\theta(d)$, $E_\theta(q)$ with $T(d)$. In this setup, it succeeds in using $E_\theta(q)$ to search over bag-of-token index $T(\mathcal{D})$. We expand on SɪDR in more detail in Appendix C.

**On-policy Retrieval.** During R3 training, we adopt the late parametric mechanism of SIDR (shwon as option 4), which first retrieves the top-$m$ documents using the bag-of-tokens index $T(\mathcal{D})$, and then embeds the these documents for re-ranking. This process can be formulated as:

$$\{\hat{d}\}_m = \mathcal{R}_\theta(E_\theta(q), T(\mathcal{D}), m)$$
$$\{\hat{d}\}_k = \mathcal{R}_\theta(E_\theta(q), E_\theta(\{\hat{d}\}_m), k)$$

To reduce training costs, we set $m = k = 20$ in our experiments. Additional ablation studies and comparisons on re-indexing solutions are provided in Section 5.1.

## 3.3 Probability-approximiated Generation

Following on-policy retrieval, we obtain the retrieved set $\{\hat{d}\}_k$ for each query, denoted as $\hat{\mathcal{D}}_q$. The goal of the LLM forward step is to divide this set into a positive pool $\hat{\mathcal{D}}_q^+$ and a negative pool $\hat{\mathcal{D}}_q^-$, based on the RAG outcome. Below, we outline the key challenges involved in this step.

**Challenge of Online Autoregressive Generation.** During training, the online generation process $\mathcal{G}$ is computationally expensive and slow due to the overhead of autoregressive decoding.

**Solution.** While a promising solution is to generate responses on-the-fly and verify them with either human-in-the-loop in the loop [56] or a larger proprietary LLM as a judge [57, 58], this approach incurs substantial computational cost, potentially slowing down training and reducing the accessibility of our method. As a more practical solution, we pre-compute a probability threshold offline and use it to approximate autoregressive generation online. This relies on the weak assumption that a higher conditional probability $P(y \mid x)$ corresponds to a greater likelihood of generating $y$, where $x$ is the input to the LLM. Further discussion on this assumption is provided in Appendix F. Below, we present the details of our solution.

**Offline Computation of Probability Threshold.** We follow the standard RAG pipeline described in Section 3.1. For each training query $q$, we retrieve the top-100 documents, denoted as the initial retrieved set $\mathcal{D}_q$. For each $d_i \in \mathcal{D}_q$, we compute the joint probability $P(y \mid x)$ of generating the ground truth response $y$ given the input prompt $x$, where $x$ consists of the query, one retrieved document, and the instruction.

$$x = \text{Prompt}_{task}(q, d_i); \qquad P(y \mid q, d_i) = P(y \mid x) = \prod_{\forall t_i \in y} P_\phi(t_i \mid t_{<i}, x)$$

We then generate responses $\hat{y} = \mathcal{G}(x)$. Based on whether $\hat{y}$ passes the evaluation, we partition the initial retrieved set $\mathcal{D}_q$ into a positive pool $\mathcal{D}_q^+$ and a negative pool $\mathcal{D}_q^-$. We set $k = 100$ and discard any query for which either pool is empty. We define the probability thresholds for positives and negatives as $\mathcal{T}^+$ and $\mathcal{T}^-$, respectively.

$$\mathcal{T}_q^+ = \max\{P(y \mid q, d) \mid \forall d \in \mathcal{D}_q^-\}; \qquad \mathcal{T}_q^- = \min\{P(y \mid q, d) \mid \forall d \in \mathcal{D}_q^+\}$$

**Online Identification of Positives and Negatives.** During online training, given a query $q$, we retrieve the top-$k$ documents as the in-training retrieved set $\hat{\mathcal{D}}_q$. For each $\hat{d}_i \in \hat{\mathcal{D}}_q$, we compute $P(y \mid q, \hat{d}_i)$ and classify it as positive or negative based on the offline thresholds.

$$\hat{d}_i \in \begin{cases} \hat{\mathcal{D}}_q^+, & \text{if } P(y \mid q, \hat{d}_i) > \mathcal{T}_q^+ \\ \hat{\mathcal{D}}_q^-, & \text{if } P(y \mid q, \hat{d}_i) < \mathcal{T}_q^- \\ \text{Discarded}, & \text{else} \end{cases}$$

In plain terms, if using $\hat{d}_i$ leads to a higher probability of generating $y$ than any $d \in \mathcal{D}_q^-$, then $\hat{d}_i$ is considered positive. Conversely, if its probability is lower than any $d \in \mathcal{D}_q^+$, then $\hat{d}_i$ is considered negative. Otherwise, $\hat{d}_i$ is discarded. If $\hat{\mathcal{D}}_q^-$ or $\hat{\mathcal{D}}_q^+$ is empty, we fall back to the initial set $\mathcal{D}_q^-$ or $\mathcal{D}_q^+$.

**Sampling.** In practice, we iterate through the retrieved documents in descending order of retrieval relevance. For each document $\hat{d}_i$, we apply the LLM forward pass to classify it as positive, negative, or discard it. This process continues until the first negative document is identified, which we designate as the hard negative $\hat{d}^-$. We then randomly sample one positive document $\hat{d}^+$ from $\hat{\mathcal{D}}_q^+$. If either

$\hat{d}^+$ or $\hat{d}^-$ is unavailable, we fall back to sampling from the initial pool $\mathcal{D}_q^+$ or $\mathcal{D}_q^-$, respectively. We evaluate alternative sampling strategies in Appendix E and find that this strategy consistently achieves the best performance.

**Other Details.** For closed-set tasks where the response is a single token, we directly compare next-token probabilities to classify $\hat{d}_i$. If the gold choice token has a higher probability than other options, $\hat{d}_i$ is considered as positive; otherwise, negative. For queries $q$ with multiple gold responses $y$, each $y$ is treated as a separate entry. If $\hat{d}_i$ succeeds on at least one, it is considered positive; otherwise negative. To avoid redundant computation, we cache all $P(y \mid q, \hat{d}_i)$ for reuse.

### 3.4 Reinforced Contrastive Learning

Following on-policy retrieval and probability-approximiated generation, we obtain the in-training retrieved positive document $\hat{d}^+$ and negative documents $\hat{d}_i^-$ for the training query $q$. In the final step, we apply contrastive learning to these pairs.

Our training objective follows SIDR to preserve its compatibility with late parametric retrieval (i.e., retrieving documents from a bag-of-tokens index and then re-ranking). Given a batch $B$ of $N$ samples, each sample contains a query $q_i$, a positive document $d_i^+$, and a negative document $d_i^-$. The semi-parametric contrastive loss is defined as:

$$L(q,d) = -\sum_{i=1}^{N}(\log \underbrace{\frac{e^{f(q_i,d_i^+)}}{\sum_{\forall d \in B} e^{f(q_i,d)}}}_{\text{q-to-d}} + \log \underbrace{\frac{e^{f(d_i^+,q_i)}}{\sum_{\forall q \in B} e^{f(d_i^+,q_i)}}}_{\text{d-to-q}})$$

The final loss integrates contrastive loss of both parametric and semi-parametric components:

$$L_{\text{para}}(q,d) = L(E_\theta(q), E_\theta(d))$$
$$L_{\text{semi-para}}(q,d) = L(E_\theta(q), T(d))/2 + L(T(q), E_\theta(d))/2$$
$$L_{\text{final}}(q,d) = L_{\text{para}}(q,d) + L_{\text{semi-para}}(q,d)$$

## 4 Experiments

### 4.1 Experimental Setup

**Tasks and Datasets.** We evaluate R3 on five public RAG benchmarks. For free-form generation, we utilize Natural Questions (NQ; Kwiatkowski et al., 2019), TriviaQA (TQA; Joshi et al., 2017), and HotpotQA [60], three well-established open-domain QA datasets. For closed-set generation, we employ the PubHealth [61] dataset for fact-checking tasks, and the ARC-Challenge [62] dataset for multiple-choice reasoning. More details can be found in Appendix A.

We exclude long-form generation datasets as we use the probability of continuation to approximate RAG performance, which may not align well with such tasks. Additionally, certain datasets, such as PopQA [63], which only offer a test split, are also excluded.

**Evaluation Metrics.** Following previous works [64, 63], we use accuracy on the test set as our evaluation metric. In the traditional IR setting, accuracy reflects whether the retrieved documents contain the answer, while in the RAG setting, it is determined by whether the generated output contains the answer. Since our training uses only one document in context, whereas prior work typically uses ten, we report accuracy under both 1-shot and 10-shot settings for comparison.

**Implementation Details.** Our RAG system employs the LLM Llama3-8b [65] with the retriever SIDR$_{\text{MS}}$ [55] that trained on MS MARCO dataset [30]. For experiments on NQ, since certain baselines are trained on its training split, we additionally train SIDR$_{\text{MS}}$ on NQ before applying our method. By defualt, we use the same English Wikipedia datastore and prompt as those open-sourced by SELF-RAG, detailed in Appendix K. For HotpotQA, we use the official datastore provided with the dataset. During training, we train the retriever for each dataset for 80 epochs, aligning with the training duration used for SIDR$_{\text{MS}}$. We use a batch size of 128 and an AdamW optimizer [66] with a learning rate of $2 \times 10^{-5}$. The training process is divided into two phases: the first half involves a warm-up phase using initial retrieved positives and negatives, while the second half transitions to in-training retrieval, using the in-training positives and negatives. During inference, we set the

maximum number of generated token to be 100 for free-form generation while 20 for closed-set generation. Our experiments are conducted with 4 NVIDIA GPUs. Both offline RAG preparation and online RAG training take less than one day, depending on the number of queries in the datasets. We leverage vLLM [67] to accelerate offline generation.

**Baselines.** We consider the below baselines: (1) *Standard RAG*: RAG frameworks using Llama3-8B and SOTA retrievers E5 [51] and CONTRIEVER$_{MS}$ [49]. (2) *Improving IR for RAG*: RAG frameworks with tuning retriever or advanced retrieval strategies enhanced by LLM. Strategy-based methods include ADAPTIVE-RAG [33] and IR-COT [35], which adapt retrieval based on question complexity or LLM reasoning, as well as DRAGIN [31], FLARE [32], and SEAKR [34], which dynamically retrieve or selectively integrate external information guided by LLMs. For learning-based method, we compare against REPLUG [40], which tunes only the retriever while frozen the LLM. (3) *Improving LLM for RAG*: RAG frameworks that tune the LLM, typically requiring more computation than tuning the retriever. We compare with SELF-RAG [64], which trains the LLM to decide when to retrieve and to self-verify its answers, and with RA-DIT [41], which jointly tunes both the retriever and the LLM. We include LLM-tuning methods in the comparison to illustrate the relative improvement from tuning the retriever. (4) *Transferring to other LLMs*: We compare the RAG framework using different LLMs, such as Llama3-Instruct$_{8B}$ [65], Phi-3-mini-4k-instruct$_{3.8B}$ [68], Mistral-Instruct$_{7B}$ [69], along with S1DR$_{MS}$ before and after tuning. This setup is designed to evaluate whether the learned in-context relevance transfers across different LLMs. Further model details are provided in Appendix B.

## 4.2 Main Results

Table 1: Main results of R3 and other RAG baselines. **Bold**: best RAG method by only improving IR. Δ: improvement or decline; ▲: baseline for below methods to compare; †: reproduction by other works; ‡: our reproduction; ♠: LLM-enhanced retrieval process.

| Task Type (→) | Free-form | | | | | | | | | Closed-set | | | | | | | |
|---|---|---|---|---|---|---|---|---|---|---|---|---|---|---|---|---|---|
| Dataset (→) | NQ | | | | TriviaQA | | | | HotpotQA | | PubHealth | | | | ARC-C | | | |
| Method (↓)  Metrics (→) | 1-shot | Δ | 10-shot | Δ | 1-shot | Δ | 10-shot | Δ | 1-shot | Δ | 1-shot | Δ | 10-shot | Δ | 1-shot | Δ | 10-shot | Δ |
| *Standard RAG* | | | | | | | | | | | | | | | | | | |
| ‡ Llama3$_{8B}$ + S1DR | 42.2 | ▲ | 42.1 | ▲ | 62.0 | ▲ | 62.5 | ▲ | 39.1 | ▲ | 63.5 | ▲ | 64.9 | ▲ | 56.9 | ▲ | 57.5 | ▲ |
| ‡ Llama3$_{8B}$ + CONTRIEVER$_{MS}$ | 36.5 | -5.7 | 38.3 | -3.8 | 60.7 | -1.3 | 60.6 | -1.9 | 37.2 | -1.9 | 63.1 | -0.4 | 62.9 | -2.0 | 58.1 | +1.2 | 58.9 | +1.4 |
| ‡ Llama3$_{8B}$ + E5 | 43.2 | +1.0 | 41.8 | -0.3 | 63.2 | +1.2 | 61.4 | -1.1 | 35.7 | -3.4 | 64.7 | +1.2 | 63.7 | -1.2 | 58.0 | +1.1 | 58.1 | +0.6 |
| *Improving IR for RAG* | | | | | | | | | | | | | | | | | | |
| ♠ADAPTIVE-RAG$_{T5-XXL-11B}$ (15-shot) [33] | – | – | 47.4 | +5.3 | – | – | 57.2 | -5.3 | 46.8 | +7.7 | – | – | – | – | – | – | – | – |
| †♠IR-COT$_{Llama-3.1-8B-Instruct}$ (15-shot) [35] | – | – | 47.8 | +5.7 | – | – | 60.8 | -1.7 | 43.8 | +4.7 | – | – | – | – | – | – | – | – |
| †♠DRAGIN$_{Llama-3.1-8B-Instruct}$ (15-shot) [31] | – | – | 48.0 | +5.9 | – | – | 66.6 | +4.1 | 43.0 | +3.9 | – | – | – | – | – | – | – | – |
| †♠SEAKR$_{Llama-3.1-8B-Instruct}$ (15-shot) [34] | – | – | 40.6 | -1.5 | – | – | 65.6 | +3.1 | 42.4 | +3.3 | – | – | – | – | – | – | – | – |
| †♠FLARE$_{Llama-3.1-8B-Instruct}$ (15-shot) [32] | – | – | 45.0 | +2.9 | – | – | 64.8 | +2.3 | 37.2 | -1.9 | – | – | – | – | – | – | – | – |
| †REPLUG$_{Llama2-7B}$ (3-shot) [70] | – | – | – | – | – | – | – | – | – | – | – | – | 41.7 | -23.2 | – | – | 47.2 | -10.3 |
| R3$_{Llama-3-8B}$ (**Ours**) | **47.8** | **+5.6** | **48.2** | **+6.1** | **65.8** | **+3.8** | 66.2 | +3.7 | **47.1** | **+8.0** | **69.5** | **+6.0** | **69.3** | **+4.4** | **59.2** | **+2.3** | **59.4** | **+1.9** |
| *Improving LLM for RAG* | | | | | | | | | | | | | | | | | | |
| Llama3-Instruct$_{8B}$ + S1DR | 47.2 | +5.0 | 54.7 | +12.6 | 65.2 | +3.2 | 73.3 | +10.8 | 49.0 | +9.9 | 67.2 | +3.7 | 71.8 | +6.9 | 72.3 | +15.2 | 75.4 | +18.0 |
| SELF-RAG$_{Llama2-7B}$ [64] | – | – | – | – | – | – | 66.4 | +3.9 | – | – | – | – | 72.4 | +7.5 | – | – | 67.3 | +9.8 |
| †SELF-RAG$_{Mistral-7B}$ [71] | – | – | – | – | – | – | 64.8 | +2.3 | – | – | – | – | 72.4 | +7.5 | – | – | 74.9 | +17.4 |
| †SELF-RAG$_{Llama-3-8B}$ [72] | – | – | – | – | – | – | 56.4 | -6.1 | – | – | – | – | 67.8 | +2.9 | – | – | 58.0 | +0.5 |
| ‡SELF-RAG$_{Llama3-8B}$ + S1DR | 38.5 | -3.7 | 38.0 | -4.1 | 51.0 | -11.0 | 57.7 | -4.8 | – | – | 64.2 | +0.7 | 64.0 | -0.9 | 58.9 | +2.0 | 59.1 | +1.6 |
| RA-DIT$_{Llama-65B}$ (5-shot) [41] | – | – | 43.9 | +1.8 | – | – | 75.1 | +12.6 | 40.7 | +1.6 | – | – | – | – | – | – | – | – |
| *Transferring R3 to other LLM* | | | | | | | | | | | | | | | | | | |
| Llama3-Instruct$_{8B}$ + S1DR | 45.2 | ▲ | 52.7 | ▲ | 65.2 | ▲ | 73.3 | ▲ | 49.0 | ▲ | 67.2 | ▲ | 71.8 | ▲ | 72.3 | ▲ | 75.4 | ▲ |
| Llama3-Instruct$_{8B}$ + R3 | 48.8 | +3.6 | 56.5 | +3.8 | 65.6 | +0.4 | 73.8 | +0.5 | 50.7 | +1.7 | 65.2 | -2.0 | 66.1 | -5.7 | 71.8 | -0.5 | 75.1 | -0.3 |
| Phi-3-mini-4k-instruct$_{3.8B}$ + S1DR | 44.1 | ▲ | 50.7 | ▲ | 64.6 | ▲ | 69.2 | ▲ | 44.1 | ▲ | 48.2 | ▲ | 57.6 | ▲ | 84.5 | ▲ | 84.7 | ▲ |
| Phi-3-mini-4k-instruct$_{3.8B}$ + R3 | 46.6 | +2.5 | 52.4 | +1.7 | 65.6 | +1.0 | 70.4 | +1.2 | 45.7 | +1.6 | 45.3 | -2.9 | 54.4 | -3.2 | 85.1 | +0.6 | 84.2 | -0.5 |
| Mistral-Instruct$_{7B}$ + S1DR | 43.9 | ▲ | 48.8 | ▲ | 58.2 | ▲ | 57.1 | ▲ | 42.4 | ▲ | 50.1 | ▲ | 57.4 | ▲ | 69.8 | ▲ | 71.2 | ▲ |
| Mistral-Instruct$_{7B}$ + R3 | 46.1 | +2.2 | 51.7 | +2.9 | 59.8 | +1.6 | 57.6 | +0.5 | 41.8 | -0.6 | 46.7 | -3.4 | 54.6 | -2.8 | 69.2 | -0.6 | 70.6 | -0.6 |

Table 1 presents results of R3 and other RAG methods, with key findings summarized below:

**Reinforced contrastive learning effectively improves the retriever within RAG environments.** Unlike CONTRIEVER$_{MS}$ and E5, which rely on extensive pre-training and SFT, R3 tunes S1DR with only four GPUs in one day. This efficient setup yields an average improvement of 5.2% over the original S1DR$_{MS}$ and outperforms other SOTA retrievers by 4.9% under the 1-shot setting. On PubHealth, the gain reaches 6%, which even instruction-tuned LLMs fail to match. These results demonstrate that learning retrieval tailored to the specific RAG environment in an RL manner is more effective than SFT with massive paired data that is unaware of the RAG environment. In Appendix I, we show that improving the retriever for RAG can degrade its performance in traditional IR settings, highlighting the gap between IR and RAG.

**Tuning a small retriever can yield improvements on par with tuning a LLM.** Reproductions of SELF-RAG from other works [72, 71] and our own experiments have shown inconsistent gains. This suggests that, despite substantial training costs, improving RAG through LLM tuning requires heavy customization and lacks generalizability. In contrast, tuning a smaller retriever can achieve comparable or even superior results to RAG-oriented or instruction-tuned 8B LLMs on certain datasets. Importantly, improving retriever complements improving LLM without conflict, offering a more efficient and controllable path for improving RAG systems.

**Learned relevance can be transferred to other LLMs across domains but not across tasks.** Our results show that R3, trained with Llama3-8B, also improves performance when paired with other LLMs, such as Llama3-Instruct-8B, Phi-3-mini-4k-instruct, and Mistral-Instruct, on QA tasks across different benchmarks. However, this transferability does not consistently extend to non-QA tasks. Given that the original SIDR is trained on a QA dataset, the current observations suggest that the learned relevance to be robust for transfer across domains within the same task, but not across tasks. This also reflects a broader phenomenon: while different LLMs may require similar documents for general QA tasks, these requirements diverge significantly for more complex tasks, such as fact-checking and complex reasoning. This highlights the importance of learning retrieval tailored to the specific RAG environment. Rather than viewing this as a limitation, we see it as evidence that R3 is capable of capturing environment-specific relevance.

## 5 Analysis

### 5.1 Ablation Study

Compared to prior works, our main differences include (i) construct constrastive on-the-fly (ii) employing contrastive objective instead of KL divergence objective, and (iii) using late parametric to avoid periodic re-indexing. We systematically analyze these factors in this section. Note that our ablation experiments on NQ are conducted with the initial retriever $SIDR_{MS}$, and thus differs from the main experiment setup.

**Contrastive Learning versus Reinforced Contrastive Learning.** In the warm-up stage, we use documents retrieved offline from the initial retrieval pool. During the training stage, we apply RCL using documents retrieved online with the current parameters (i.e., on-policy). We conduct an ablation study on NQ and PubHealth under three settings: *[offline-only]* uses only the offline retrieved pool offline, which can be considered as conventional CL; *[online-only]* skips the warm-up and relies solely on online on-policy retrieval; our full method *[offline+online]* uses both offline warm-up and online on-policy retrieval, which refers to RCL with both positive and negative pools guaranteed.

As shown in Figure 4, using either *[offline-only]* or *[online-only]* alone results in suboptimal performance, while *[offline+online]* leads to the best results. These observations demonstrate that both off-policy contrastive labels obtained through conventional CL and on-policy contrastive labels identified via RCL during training can improve the retriever. However, combining both approaches delivers the best results. This supports findings from our prior work [55], which show that a warm-up phase with static contrastive labels helps the retriever initially align with environment-specific relevance, and on-policy retrieval further refines it by continuously mining more effective contrastive labels during training, thereby deepening the retriever's alignment with the environment.

**Contrastive Objective versus KL-Divergence Objective.** We also evaluate a variant of our method that replaces contrastive learning objective with KL divergence, denoted as *[offline+online(KL)]*. KL divergence has been adopted in prior work [40, 2] to align query-document relevance with generation likelihood. Our results show that while KL divergence yields initial improvements, these gains quickly plateau and remain consistently lower than those achieved by our contrastive approach. KL-based alignment is easier to implement but provides a more rigid and less adaptive learning signal. In contrast, contrastive learning continuously supplies effective positives and negatives based on the current retriever state, enabling ongoing improvement throughout training.

We believe KL divergence underperforms in this context because the generation likelihood distribution of LLMs is inherently hard to capture by bi-encoder. Many studies in knowledge distillation [73, 74] have attempted to use small bi-encoders to approximate fine-grained ranking signals, typically provided by cross-encoders or LLMs, but have consistently found that such models struggle to capture these signals effectively. This limitation has led to the widespread adoption of the retrieve-then-rerank paradigm in both academic and industrial settings.

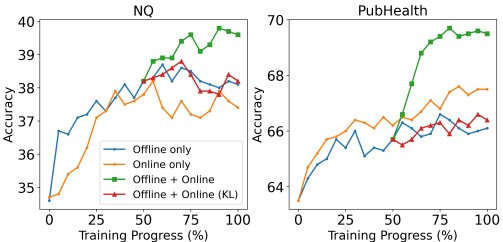
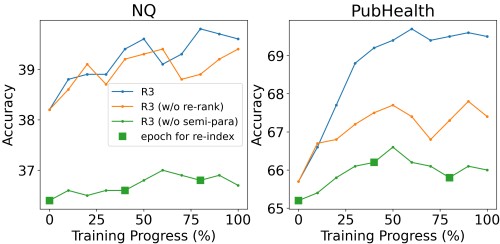

Figure 4: Ablation of offline and online learning.    Figure 5: Ablation on re-indexing strategies.

**Late Parametric versus Periodic Re-indexing.** Another key distinction between our method and prior approaches lies in how we address index staleness. While previous works periodically rebuild the index during training, we adopt a late parametric mechanism from SɪDR to avoid frequent re-indexing. Below, we compare these strategies for handling index staleness. *[R3 (w/o re-rank)]* refers to performing retrieval directly on the bag-of-tokens index without subsequent re-ranking. This reduces computational cost but results in suboptimal retrieval accuracy. *[R3 (w/o semi-para)]* denotes the removal of the entire semi-parametric design, falling back to periodic re-indexing as done in prior work. In our setup, the index is rebuilt three times during training. More details are provided in Appendix H.

As shown in Figure 5, our results indicate that while periodic re-indexing R3 *(w/o semi-para)* provides some improvement, it is significantly less effective than the semi-parametric design. The late parametric approach not only yields better performance but also reduces training cost and simplifies implementation. Compared to direct retrieval on the bag-of-tokens index, the late parametric design provides higher-quality positive and negative examples to improve. This advantage is particularly evident on the PubHealth dataset, where R3 shows a much larger improvement over R3 *(w/o re-rank)*.

## 5.2  Cost-Effectiveness Analysis

The main training cost comes from LLM forward passes to compute probability $P(y \mid q, d)$ for previously unseen documents. In Table 2, we report the number of such documents processed during training. Each query involves between 14 and 128 unseen documents, depending on the task. We observe a positive correlation between the number of processed documents and the performance gain of R3. Notably, PubHealth encounters most unseen documents and yields the largest improvement, suggesting a greater gap between the initial and learned retriever.

|         | NQ    | TriviaQA | PubHealth | ARC   |
|---------|-------|----------|-----------|-------|
| **nDoc**  | 34    | 18       | 128       | 14    |
| **Improv.** | +5.6% | +3.8%    | +6.0%     | +2.3% |

Table 2: Number of documents requiring probability computation and corresponding improvement.

Together with the transferability results in Table 1, this motivates further analysis. In Appendix G, we show that R3 learns LLM-specific relevance on PubHealth. We find that transferability improves initially but then declines steadily during training, even as RAG performance continues to rise. This suggests that the retriever becomes more specialized for the target LLM (Llama3-8B), reducing its generalization to other LLMs. This pattern reflects a broader trend: while LLMs may share similar relevance preferences for general QA tasks, they might diverge in specialized domains and complex tasks, highlighting the need to tune retrievers for each specific RAG enviroment.

## 6  Conclusion

In this work, we demonstrate the relevance gap between IR and RAG. To address this gap, we introduce R3, a RAG framework that optimizes retrieval for a given RAG environment in a RL manner. Our results show that RCL, which constructs on-policy contrastive signals on-the-fly, significantly improves a retriever within a specific RAG environment, outperforming state-of-the-art off-the-shelf retrievers and LLM-augmented retrieval, while performing comparably to RAG systems with post-trained or instruction-tuned 8B LLMs. These findings highlight the substantial potential of our approach to enhance RAG systems through effective retrieval learning.

## Limitations

Our work explores optimizing retrieval for RAG in a reinforcement learning manner, but several limitations remain. First, the current experimental setup focuses on a vanilla RAG workflow, where only one document is retrieved as context. We believe this idea can naturally extend to more complex workflows, as long as the online learning process remains computationally feasible. Second, the current reward function is constrained by its reliance on short answers and string matching. During training, we also observe that performance is sensitive to how positive and negative examples are defined. A more stable and generalizable reward or evaluation function, such as using an LLM-as-a-Judge, could further improve reliability. Lastly, our current foundation model is based on BERT, which is relatively outdated. We believe that incorporating more advanced LLM-based encoders and tuning them in a semi-parametric manner could greatly broaden the applicability and user-facing potential of this framework.

## Broader Impact and Future Work

In this section, we discuss the broader societal impact of our work and its potential influence on other areas and advanced applications.

**RAG Application Safety.** Our work could be deployed in complex environments where retrieval is learned to affect AI behavior and safety. The retrieval module can learn to retrieve contextual information, few-shot examples, instructions, or existing chain-of-thought prompts that guide LLMs toward more reliable outputs, or conversely be intentionally misused to generate harmful content, mislead users, or surface inappropriate material. We acknowledge this potential for malicious use and encourage responsible deployment and safeguard mechanisms in practice.

**Scaling with Learned Sparity.** The success of different areas can be unified under the paradigm of *Scaling with Learned Sparsity*. For example, LLM pre-training leverages Mixture-of-Experts (MoE) to scale up the parameter space, improving effectiveness while sparsely activating modules at inference time to ensure efficiency. RAG can be viewed as scaling knowledge coverage by integrating external databases, while sparsely activating specific data partitions through the retrieval function. Similarly, a line of sparse retrieval studies [75, 76, 77] can be regarded as scaling the representation space to enhance representational capacity and allivate the limitations [78] of embedding models, while sparsely activating dimensions to guarantee search efficiency.

|  | MoE | RAG | Sparse Retrieval |
|---|---|---|---|
| Area | Pre-training | Inference | Representation Learning |
| Scaling | Parameter Space | Knowledge Space | Representation Space |
| Sparsely Activate | Neural Modules | Data Partitions | Dimensions |

Table 3: Overview of scaling with learned sparsity paradigms.

Looking forward, we believe that future AI systems will inevitably require learning to scale along certain axes while simultaneously learning sparsity over them. Our two works, SIDR and R3, demonstrate how to learn sparsity and how such sparsity can further be utilized to improve the large-scale complex systems within which they operate, offering preliminary insights in this direction.

**Multi-modal RAG.** Beyond document retrieval, emerging AI search systems increasingly involve multimodal data such as images and videos. Unlike query–document pairs, where lexical relevance can often serve as a reasonable heuristic, constructing effective relevance pairs for cross-modal data is far more challenging due to the high cost of annotation and the lack of explicit alignment signals. This direction highlights opportunities for framework that can automatically obtain contrastive labels through interaction or feedback signals, a principle that aligns with the learning dynamics explored in our work.

**Searching Tools, Skills, MCP, and Agents.** Recent progress in agentic frameworks, where AI agents learn to call tools, skills, MCP services, or interact with other agents, has largely relied on online reinforcement learning to manage a small pool of available operations. As the pool of callable components continues to grow, learning effective representations of these components for retrieval will become crucial for scalable and efficient decision-making. In such settings, the proposed reinforced contrastive learning offers one possible direction for enabling search and coordination across a large pool of tools, skills, and agents.

**Acknowledgments**

Lei Chen's work is partially supported by National Key Research and Development Program of China Grant No. 2023YFF0725100, National Science Foundation of China (NSFC) under Grant No. U22B2060, Guangdong-Hong Kong Technology Innovation Joint Funding Scheme Project No. 2024A0505040012, the Hong Kong RGC GRF Project 16213620, RIF Project R6020-19, AOE Project AoE/E-603/18, Theme-based project TRS T41-603/20R, CRF Project C2004-21G, Key Areas Special Project of Guangdong Provincial Universities 2024ZDZX1006, Guangdong Province Science and Technology Plan Project 2023A0505030011, Guangzhou municipality big data intelligence key lab, 2023A03J0012, Hong Kong ITC ITF grants MHX/078/21 and PRP/004/22FX, Zhujiang scholar program 2021JC02X170, Microsoft Research Asia Collaborative Research Grant, HKUST-Webank joint research lab and 2023 HKUST Shenzhen-Hong Kong Collaborative Innovation Institute Green Sustainability Special Fund from Shui On Xintiandi and the InnoSpace GBA.

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

## A  Details of Datasets

We present details of datasets as follows.

- Natural Questions (NQ; Kwiatkowski et al., 2019) is a widely used open-domain QA dataset constructed from Wikipedia. The questions originate from Google search queries, and the answers are text spans within Wikipedia passages. This dataset consists of queries with one or more answer strings, requiring RAG systems to generate responses based on factual knowledge.
- TriviaQA (TQA; Joshi et al., 2017) is a challenging QA dataset that comprises question-answer pairs curated by trivia enthusiasts along with independently gathered evidence documents.
- HotpotQA [60] is a multi-hop question answering dataset that requires reasoning over multiple Wikipedia paragraphs to answer factoid questions.
- PubHealth [61] is a fact-checking task that focuses on verifying health claims across a variety of biomedical topics.
- ARC-Challenge [62] is a multiple-choice reasoning dataset consisting of science exam questions for grades 3 to 9.

# B Details of Models

## B.1 Retrieval Model (IR)

- E5 [51] is a state-of-the-art dense retriever that pre-trained on millions of weakly related text pairs from the Web. The unsupervised version of this model is denoted as E5-unsup. This model undergoes further fine-tuning on natural language inference (NLI) datasets, as well as the Natural Questions and MS MARCO datasets, to enhance its capabilities in downstream applications. The fine-tuned version is denoted as E5.

- CONTRIEVER [49] is a widely-used dense retriever pre-trained unsupervised on Wikipedia data and CCNet [79]. The unsupervised version of this model is denoted as CONTRIEVER. It is further fine-tuned on the MS MARCO dataset to enhance its retrieval performance, with the fine-tuned version denoted as $\text{CONTRIEVER}_{MS}$.

- DPR [28] is a widely used dense passage retriever initialized with a BERT-based uncased encoder [80], and fine-tuned on downstream dataset. Specifically, $\text{DPR}_{MS}$ is fine-tuned on the MS MARCO dataset, $\text{DPR}_{NQ}$ on the NQ dataset, and $\text{DPR}_{TQA}$ on the TriviaQA dataset.

- SIDR [55] is a semi-parametric sparse retriever that supports using both embeddings and tokenization as index. This nature allows for in-training retrieval, where the model's parameters dynamically update while the retrieval index remains fixed. The model is initialized with a BERT-based uncased encoder [80] and fine-tuned exclusively on single dataset depending on the variant: $\text{SIDR}_{MS}$ is fine-tuned on the MS MARCO dataset, $\text{SIDR}_{NQ}$ on the NQ dataset, and $\text{SIDR}_{TQA}$ on the TriviaQA dataset.

All the above retrieval methods are initialized with a BERT-based encoder, which contains approximately 200 million (0.2B) parameters.

## B.2 Large Language Model (LLM)

- $\text{Llama3}_{8B}$ [65] is a variant of the latest Llama3 model series with 8 billion parameters.

- $\text{Llama3-Instruct}_{8B}$ [65] builds upon the $\text{Llama3}_{8B}$ by undergoing a post-training stage in which the model is specifically tuned to follow instructions and align with human preferences to improve specific capabilities.

- $\text{Phi-3-mini-4k-instruct}_{3.8B}$ [68] is a lightweight widely-used LLM with 3.8 billion parameters, trained on the Phi-3 dataset featuring synthetic and high-quality filtered web data, focused on reasoning and quality.

- $\text{Mistral-Instruct}_{7B}$ [69]. We use Mistral-7B-Instruct-v0.3 LLM which is an instruct fine-tuned version of the Mistral-7B-v0.3.

## B.3 Retrieval-augmented Generation Framework (RAG)

- ADAPTIVE-RAG [33] is a RAG framework that adapts the retrieval strategy based on question complexity where simple questions are answered without retrieval and complex ones trigger a multi-step process involving iterative interaction between the retriever and the LLM.

- IR-CoT [35] is a RAG framework designed for multi-hop question answering. It interleaves retrieval with intermediate steps in chain-of-thought (CoT) reasoning.

- FLARE [32] is a RAG framework that enhances LLM performance by selectively integrating external knowledge. It monitors token-level generation probabilities to detect uncertainty and triggers retrieval when needed during generation.

- DRAGIN [31] is a RAG framework that, similar to FLARE, monitors token-level probabilities during generation to detect uncertainty.

- SEAKR [34] is a RAG framework that reduces hallucinations by leveraging self-aware uncertainty. It monitors the LLM's internal signals and triggers retrieval only when high uncertainty is detected during generation.

  For the above four baselines, we adopt the reproductions and reported results from Moskvoretskii et al. [19].

- REPLUG [40] is a RAG framework using GPT-3 and CONTRIEVER. The retriever is specifically trained to use the first 128 tokens of a sequence as queries, with the goal of retrieving documents that maximize the probability of generating the subsequent 128 tokens when these retrieved documents are prepended to the query.

- SELF-RAG [64] is a RAG framework designed to improve response quality by enabling on-demand retrieval and incorporating self-reflection mechanisms.

  The reproductions by Wang et al. [71] and Zhang et al. [72], SELF-RAG$_{\text{Mistral-7B}}$ and SELF-RAG$_{\text{Llama3-8B}}$ respectively, involve tuning Mistral-7B and Llama3-8B as base language models using the open-source data provided by SELF-RAG. Our reproduction, SELF-RAG$_{\text{Llama3-8B}}$+SIDR$_{\text{MS}}$, utilizes the SELF-RAG$_{\text{Llama3-8B}}$ checkpoint from Zhang et al. [72] as LLM, while employing the same retriever SIDR$_{\text{MS}}$ and adapting it to our downstream setup.

- RA-DIT [41] is a RAG framework that separately fine-tunes the LLM to better utilize retrieved information and the retriever to align with the LLM's preferences.

## C   Revisiting Semi-parametric Disentangled Retriever (SIDR)

This section can be considered an expanded version of Section 3.2, where we discuss SIDR in more detail.

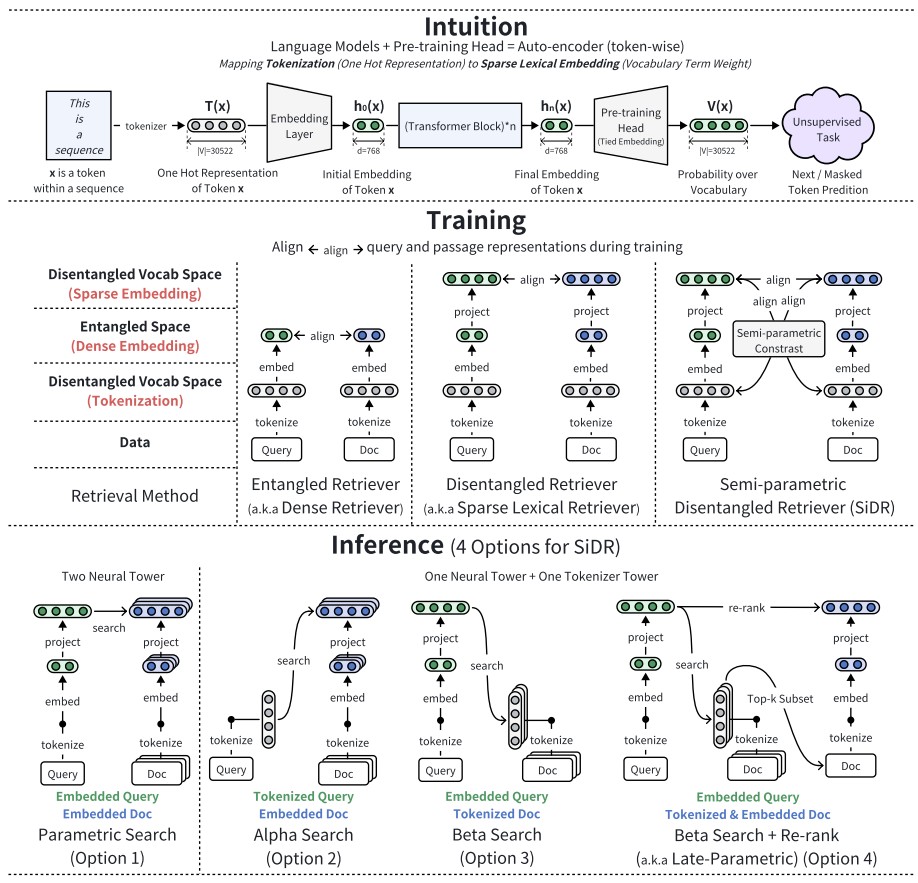

Figure 6: Overview illustration of SIDR.

**Background: Challenge of Index Staleness.** As is well known, the retrieval index is typically the embedding of the datastore, denoted as $E_\theta(\mathcal{D})$. During training, as the $\mathcal{R}_\theta$ parameters update from $\theta \to \theta'$, the index must be updated accordingly, from $E_\theta(\mathcal{D}) \to E_{\theta'}(\mathcal{D})$, where $|\mathcal{D}| \gg 1$. Otherwise, the index becomes stale, leading to mismatches between the query and document

embeddings. However, frequently re-embedding a large datastore is computationally expensive and often unaffordable in practice. Fundamentally, this challenge stems from the coupling between retrieval index and model parameters $\theta$.

**Intuition of SIDR.** Intuitively, a language model (LM), together with its pre-training head, can be viewed as an auto-encoder. Let's use BERT as an example. Given an input sequence $X$, a specific token $x$ can be represented as a one-hot vector $T(x) \in {0, 1}^{|V|}$, where $|V| = 30522$ is the size of the BERT vocabulary $V$. These tokens are then processed by the embedding layer to yield a $d$-dimensional hidden state $H_0(x)$, where $d = 768$. After passing through 12 transformer blocks, the token representations interact with the contextualized token representations, resulting in the final token representation from the last layer, denoted as $H_n(x)$. The $H_n(x)$ of a masked token is then further processed by the masked language model head (MLMH) to yield a probability distribution $V(x) \in (0, 1)^{|V|}$ over the vocabulary space, used to predict the masked token.

Given a sequence $X$, SIDR embeds it into a $|V|$-dimensional sparse embedding, denoted as $E_\theta(X) \in (0, +\infty)^{|V|}$. SIDR builds upon a BERT-based model with three simple modifications to its pre-training head: (i) replacing the softmax activation with elu1p; (ii) aggregating token representations via max pooling; and (iii) applying top-$k$ sparsification.

Inspired by the auto-encoder property, we are curious whether output sparse embedding can be aligned with input tokenization to yield better features. Specifically, during training, SIDR applies contrastive learning across three channels: $E_\theta(q)$ with $E_\theta(d)$, $T(q)$ with $E_\theta(d)$, and $E_\theta(q)$ with $T(d)$. In this training setup, we find that SIDR successfully uses $E_\theta(q)$ to search over the bag-of-token index $T(\mathcal{D})$, as illustrated in Figure 6 with Option 3 (Beta Search) and Option 4 (Beta Search + Re-rank). This provides an opportunity to use a non-parametric bag-of-token index as a frozen proxy for retrieval during the retriever's training loop, thereby bypassing the re-indexing issue.

**On-policy Retrieval.** During R3 training, we adopt the late parametric mechanism of SIDR (shwon as option 4), which first retrieves the top-$m$ documents using the bag-of-tokens index $T(\mathcal{D})$, and then embeds the these documents for re-ranking. This process can be formulated as:

$$\{\hat{d}\}_m = \mathcal{R}_\theta(E_\theta(q), T(\mathcal{D}), m)$$
$$\{\hat{d}\}_k = \mathcal{R}_\theta(E_\theta(q), E_\theta(\{\hat{d}\}_m), k)$$

To reduce training costs, we set $m = k = 20$ in our experiments.

**Other Applications of SIDR.** Besides using the bag-of-token index to enable on-policy training in R3 (i.e., $E_\theta(q)$ searches over $T(\mathcal{D})$), SIDR offers additional advantages. For example, it facilitates conventional search, where the sparse embedding of the query searches over the sparse embeddings of the documents, enabling efficient search via inverted indexing and improving interpretability. Additionally, it can further reduce online query embedding time by using the bag-of-token representation of the query (i.e., $T(q)$ searches over $E_\theta(\mathcal{D})$), a method known as alpha search [81].

# D   Preliminary Study

In Table 4, we report the performance of several off-the-shelf retrievers across multiple datasets, covering both IR and RAG scenarios. The datasets include QA benchmarks such as NQ and TQA, and non-QA benchmarks such as PubHealth and ARC, as detailed in Appendix A. The evaluated retrievers include the dense retriever DPR, the sparse retriever SIDR, and unsupervised models such as CONTRIEVER and E5-unsup, along with their supervised counterpart. Model details are provided in Appendix B. The evaluation metric is described in Section 4.1, where the IR measure evaluates whether the retrieved document contains the answer, and the RAG measure evaluates whether the retrieved document can prompt the LLM to generate the answer. We benchmark all models using $SIDR_{MS}$ as the base retriever, which also serves as the backbone of our proposed method. The results reveal two key findings.

**Finding 1: Better IR performance does not always lead to better RAG performance, especially when a task shift occurs.** For QA tasks, we observe that higher accuracy in IR settings generally correlates with better RAG performance. However, this trend does not hold for non-QA tasks. For example, on PubHealth, a vanilla $DPR_{MS}$ outperforms a SOTA retriever in RAG settings, despite being weaker in IR metrics. Similarly, on ARC, the unsupervised CONTRIEVER achieves the best RAG performance, outperforming its supervised counterparts.

Table 4: Accuracy in IR and RAG settings using Llama3-8b with top-1 retrieved document in-context; **Bold**: best performance; Δ: improvement or decline compared to SIDR$_{MS}$; §: has been trained in-domain.

| Dataset (→) | NQ | | | | TriviaQA | | | | PubHealth | | ARC-C | |
|---|---|---|---|---|---|---|---|---|---|---|---|---|
| Retriever (↓) | *IR* | Δ | *RAG* | Δ | *IR* | Δ | *RAG* | Δ | *RAG* | Δ | *RAG* | Δ |
| *Unsupervised Pre-training* | | | | | | | | | | | | |
| Contriever | 23.6 | -15.5 | 30.9 | -3.5 | 37.2 | -18.9 | 56.6 | -5.4 | 61.8 | -1.7 | **58.6** | +1.7 |
| E5-unsup | 30.8 | -8.3 | 33.4 | -1.0 | 39.5 | -16.6 | 54.3 | -7.7 | 62.9 | -0.6 | 58.3 | +1.4 |
| *Supervised on MSMARCO* | | | | | | | | | | | | |
| DPR$_{MS}$ | 38.9 | -0.2 | 34.9 | +0.5 | 43.7 | -12.4 | 55.2 | -6.8 | 64.5 | +1.0 | 56.3 | -0.6 |
| SiDR$_{MS}$ | 39.1 | – | 34.4 | – | 56.1 | – | 62.0 | – | 63.5 | – | 56.9 | – |
| *Supervised on NQ* | | | | | | | | | | | | |
| DPR$_{NQ}$ | ‡43.5 | +4.4 | ‡38.5 | +4.1 | 39.4 | -16.7 | 55.9 | -6.1 | 62.9 | -0.6 | 56.6 | -0.3 |
| SiDR$_{NQ}$ | ‡49.5 | +10.4 | ‡42.7 | +8.3 | 47.4 | -8.7 | 59.8 | -2.2 | 63.5 | – | 57.1 | +0.2 |
| *Supervised on TQA* | | | | | | | | | | | | |
| DPR$_{TQA}$ | 32.1 | -7.0 | 32.9 | -1.5 | ‡55.4 | -0.7 | ‡61.1 | -0.9 | 63.1 | -0.4 | 56.7 | -0.2 |
| SiDR$_{TQA}$ | 30.6 | -8.5 | 32.9 | -1.5 | ‡56.9 | +0.8 | ‡**63.6** | +1.6 | 61.1 | -2.4 | **58.6** | +1.7 |
| *Pre-training + Supervised on Multiple Datasets* | | | | | | | | | | | | |
| Contriever$_{MS}$ | 41.5 | +2.4 | 36.5 | +2.1 | 53.5 | -2.6 | 60.7 | -1.3 | 63.1 | -0.4 | 58.1 | +1.2 |
| E5 | ‡**58.0** | +18.9 | ‡**43.2** | +8.8 | **58.7** | +2.6 | 63.2 | +1.2 | **64.7** | +1.2 | 58.0 | +1.1 |
| **Upper-bound** | 65.9 | | 77.6 | | 78.5 | | 80.3 | | 92.1 | | 71.5 | |

**Finding 2: Substantial room for improvement in retrieval within the RAG system.** We find that although most required knowledge already exists in the datastore, each retriever succeeds on only a subset of queries. For example, on NQ, 77% of queries can be answered by at least one retriever via RAG, yet the best single retriever covers only 43%. This gap reveals significant untapped potential in the datastore and underscores the need for stronger retrieval in RAG.

## E   Sampling Strategies

We report the performance using SIDR$_{MS}$ as the initial retriever under the 1-shot setting. The sampling strategy plays a crucial role in forming effective contrastive pairs during training. In this section, we compare our current sampling method against two alternatives:

- **Current (A)**: Most-relevant negative + randomly selected positive
- **Alternative (B)**: Randomly selected negative + randomly selected positive
- **Alternative (C)**: Most-relevant negative + most-relevant positive

Table 5 reports the RAG performance on NQ and PubHealth under each strategy.

Table 5: Impact of sampling strategies on RAG performance using SIDR$_{MS}$ as the initial retriever.

| Strategy | NQ | PubHealth |
|---|---|---|
| A | 39.8 | 69.5 |
| B | 38.7 | 68.7 |
| C | 39.2 | 69.0 |

Overall, the current strategy (A) achieves the best performance on both datasets. We hypothesize that incorporating the most-relevant negative increases training difficulty and encourages more discriminative learning, while introducing randomness in positive sampling promotes generalization. These findings are consistent with observations from prior retrieval learning studies (e.g., DPR), where the choice of hard negatives has a substantial impact on performance, whereas positive selection tends to be less critical.

## F   Assumption on Probability and Generation

Given that our training relies on the conditional probability of the ground-truth response $y$ given input $x$, denoted as $P(y \mid x)$, as a signal for identifying positive and negative documents, we now

Table 6: Results of RAG framework using top-1 and top-10 documents in context, sorted by retrieval relevance and joint probability of responses.

| Task Type ($\rightarrow$) | | Free-form | | | | Closed-set | | | |
|---|---|---|---|---|---|---|---|---|---|
| Dataset ($\rightarrow$) | | NQ | | TriviaQA | | PubHealth | | ARC-C | |
| Method ($\downarrow$)   Metrics ($\rightarrow$) | 1-shot | 10-shot | 1-shot | 10-shot | 1-shot | 10-shot | 1-shot | 10-shot |
| Llama3$_{8B}$ + S$_I$DR$_{MS}$ (doc with top relevance) | 49.1 | 51.4 | 65.3 | 67.2 | 65.2 | 67.4 | 58.1 | 57.3 |
| Llama3$_{8B}$ + S$_I$DR$_{MS}$ (doc with top $P(y \mid x)$) | 85.1 | 76.2 | 88.7 | 84.2 | 87.4 | 77.4 | 95.6 | 83.6 |

investigate whether higher values of $P(y \mid x)$ correlate with improved RAG performance (i.e., better generation of $y$).

For each dataset, we sample 1,000 training instances. For each query, we retrieve the top 100 documents, and evaluate RAG performance using only the top-1 and top-10 documents, selected either by retrieval relevance or by $P(y \mid x)$. The results, presented in Table 6, show that $P(y \mid x)$ is predictive of final RAG accuracy. Moreover, the strong performance obtained when using documents ranked by $P(y \mid x)$ highlights untapped potential of the datastore, a potential that current retrieval methods fail to fully leverage.

To our knowledge, using $P(y \mid x)$ to identify positives and negatives is a rough yet resource-efficient solution that could cover most existing knowledge-intensive tasks, aligning with their evaluation metrics that often utilize string matching. However, it may not be suitable for long-form generation, which requires different evaluation strategies. We believe it is possible to customize the identification of positive and negative examples based on the specific needs of each task. Ideally, if computational cost is not a concern or resources are sufficient, a strong proprietary LLM like GPT-4 can be used for identification on-the-fly.

# G   LLM-Specific Relevance

In Table 2, we observe that PubHealth requires significantly more online probability computations and yields larger improvements compared to other datasets. Combined with the transferability results in Table 1, we hypothesize that R3 may succeed in learning LLM-specific relevance on this task. While this enables strong task-specific performance, it also limits the retriever's generalization ability across LLMs. To further examine this hypothesis, we report both effectiveness and transferability metrics throughout the online training process of R3 on PubHealth.

Table 7: Effectiveness and transferability during different stages of online training on PubHealth.

| On-policy Training Progress | 0% | 25% | 50% | 75% | 100% |
|---|---|---|---|---|---|
| **RAG Effectiveness** | 65.7 | 67.7 | 68.8 | 69.4 | 69.5 |
| **Transfer to Llama3-8B-Instruct** | 66.7 | 67.1 | 67.0 | 66.1 | 65.2 |
| **Transfer to Mistral-7B-Instruct** | 49.8 | 50.4 | 48.2 | 47.7 | 46.7 |

We observe that while transferability initially increases, it steadily declines as training progresses, even as task-specific RAG performance continues to improve. This trend indicates that the retriever is learning a relevance function increasingly specialized for the target LLM. Although this improves retrieval quality for the specific LLM, it comes at the cost of generalization to others. This reflects a broader phenomenon: while LLMs may share similar relevance preferences for general tasks such as QA, their needs diverge more significantly in specialized domains like fact-checking. These results underscore the value of end-to-end retrieval learning for RAG.

# H   Late Parametric vs. Periodic Re-indexing

A key distinction between our work and prior practices lies in our use of the late parametric mechanism to avoid re-indexing during training. In this section, we systematically evaluate these in-training retrieval approaches.

**Baseline.** We present ablation studies on different in-training retrieval approaches: (i) R3 employs the late parametric method as proposed in S$_I$DR, which uses a bag-of-token index for first-stage retrieval

and re-ranks the top-20 documents on-the-fly using up-to-date parameters. (ii) R3 (w/o re-rank) employs the bag-of-token index for retrieval, similar to the late parametric method but without the re-ranking process. This setup aims to assess the costs associated with re-ranking during training. (iii) R3 (w/o semi-para) involves periodic re-indexing using the most recently built but outdated index for retrieval, an in-training retrieval method that commonly used in prior studies. In this setup, we employ $DPR_{MS}$ as the initial retriever. We avoid using $SIDR_{MS}$, which has high-dimensional embeddings of 30,522, in stark contrast to DPR's 768 dimensions. This significant discrepancy prevents our GPU cards from allocating the parametric index for $SIDR_{MS}$, although they manage DPR effectively.

**Training.** All models undergo the similar training pipeline: they are trained for 80 epochs with the first 40 epochs as a warm-up and the last 40 conducting in-training retrieval. They differ only in their in-training retrieval strategies: both R3 and R3 (w/o re-rank) do not require re-indexing; R3 (w/o semi-para) requires rebuilding index at every 15 epochs (around 5k steps), a rebuild interval commonly used in previous research [54], resulting in a total of three rebuilds.

**Results.** We present the RAG accuracy on NQ and PubHealth test splits during in-training retrieval, with results reported every four epochs, as depicted in Figure 7. For the re-ranking setup, significant improvements are observed in the PubHealth data when re-ranking is employed, whereas the NQ dataset shows only minor improvements. Given that the costs associated with re-ranking are manageable in our setup, we continue to implement it. Regarding re-indexing, our findings indicate that despite requiring significant time and resources, it fails to yield improvements comparable to those of the late parametric approach and significantly lags behind. We attribute this to index staleness, where query embeddings must optimize against outdated document embeddings, rendering the learning process less effective. On the other hand, as presented in the study by Zhou et al. [55], by re-ranking the top-20 retrieved documents, the late parametric method can recover more than 90% of the performance of a full parametric search across different tasks, representing a minor compromise. This also partially explains why the late parametric approach outperforms periodic re-indexing.

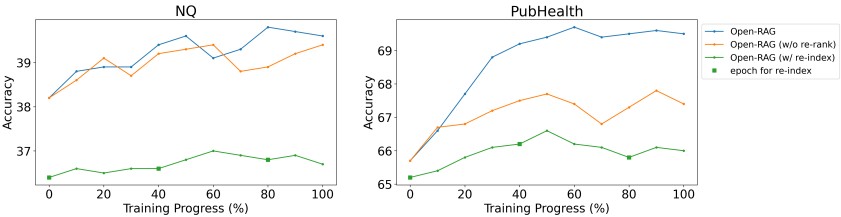

Figure 7: RAG accuracy of different in-training retrieval approaches.

# I   Inconsistencies between IR and RAG Settings

## I.1   Performance Changes in IR Scenarios after Tuning

Table 8: Performance changes before and after tuning the retriever using the R3 approach.

| Dataset (→) | | NQ | | TriviaQA | |
|---|---|---|---|---|---|
| Method (↓)     Metrics (→) | IR | RAG | IR | RAG |
| $Llama3_{8B}$ + $SIDR_{MS}$ | 39.1 | 34.4 | 56.1 | 62.0 |
| $Llama3_{8B}$ + R3 ($SIDR_{MS}$) | 40.8 (+1.7) | 39.8 (+5.4) | 53.9 (-2.2) | 65.8 (+3.8) |
| $Llama3_{8B}$ + $SIDR_{NQ}$ | 49.5 | 42.7 | – | – |
| $Llama3_{8B}$ + R3 ($SIDR_{NQ}$) | 47.1 (-2.4) | 44.1 (+1.4) | – | – |

We evaluate the performance of our retriever in both IR and RAG scenarios before and after tuning. Note that the retriever we use for NQ is $SIDR_{MS}$, which does not access the NQ training split, unlike the version used in the main experiments. In IR scenarios, we measure top-1 retrieval accuracy by checking whether the top-1 retrieved document contains the answer. In RAG scenarios, we measure accuracy using a single document in the context window, evaluating whether the generated response contains the correct answer.

Our results indicate that while R3 tunes the retriever to improve RAG performance, it results in inconsistent performance on traditional IR performance, with some degradation observed on certain datasets. This highlights a long-standing issue in the IR evaluation pipeline: a document containing the answer does not necessarily imply that it effectively addresses the query, and conversely, a document not containing the answer does not mean it is irrelevant or unhelpful.

Our conclusion also aligns with the findings and observations of other research. Cuconasu et al. [23] find that including more answer-containing documents in the context negatively impacts RAG performance. Similarly, Nian et al. [82] observe that traditional relevance definitions for IR tasks do not enhance RAG response quality. Additional research emphasizes the need for further learning to bridge the preference gap [83] or re-ranking [84] for off-the-shelf retrievers to improve RAG performance.

## I.2  Case Study

In this section, we present a case study using the NQ dataset where each query has a list of answer strings. This case study is designed to further explore the inconsistency issues inherent in RAG implementations. We specifically examine two scenarios: (i) cases where the retrieved document contains the correct answer but fails to produce the correct RAG output, and (ii) instances where the retrieved document does not directly address the query, yet the RAG model manages to generate the correct answer nonetheless. To enhance our analysis, we also ask GPT-4 to judge whether the documents address the question, helping readers quickly grasp the key issue.

In Figure 8, we present examples where RAG outputs the correct answer, even though the retrieved document neither contains the answer nor is considered to address the question by GPT-4. In both cases, the document fails to provide the correct answer or relevant clues, yet RAG is still able to generate the correct response. We believe this is a common phenomenon, as LLMs possess a wealth of internal knowledge, particularly for public knowledge questions. In general, an incorrect or imperfect retrieved document is insufficient to mislead the LLM into producing an incorrect output.

In Figure 9, we present examples where RAG fails to output the correct answer, even though the retrieved document contains the correct answer or GPT-4 considers the document as addressing the question. In the first case, the document does not address the query, and the LLM tends to extract key phrases, such as the title, as the response, ignoring the query. In the second case, the document contains information that addresses the query, and the LLM generates the correct answer, but the answer's alias name is not included in the pre-defined answer candidates, leading to a failure in the RAG scenario. These inconsistencies can be driven by many factors, including the LLM, instruction prompt, evaluation metrics, and relevance. All of these factors are intertwined, and we believe that end-to-end data-driven learning is more effective than analyzing their interplay in isolation.

# J  Examples of Generation

For free-form generation tasks, we assess whether the generation contains any of the given answers. For closed-set generation tasks, we measure whether the generation contains the answers. Below are examples that illustrate how different generations pass (1) or fail (0) the evaluation.

# K  Prompt Formats

We demonstrate our prompts for different tasks such as OpenQA, fact-checking, and multi-choice reasoning in Figures 12, 13, and 14, respectively.

```
==================================== Question ====================================
Who plays Big Momma in Big Momma's House?
==================================== Answers ====================================
 ['Ella Mitchell', 'Martin Lawrence']
==================================== Document ====================================
Bounce with Me
Jermaine Dupri, Jagged Edge and Da Brat. Brief clips from "Big Momma's House" are also
included. Bounce with Me "Bounce with Me" is a single by American rapper Lil' Bow Wow
featuring Xscape. It is Lil' Bow Wow's debut single, taken from his debut album "Beware of
Dog". The single samples "Love Serenade (Part II)" by Barry White. It spent nine weeks at
number 1 on the U.S. Rap Chart and number 20 on "Billboard" Hot 100. The song was featured
in the film "Big Momma's House" and the trailer of "Hardball".
==================================== Generation ====================================
[1] Martin Lawrence
==================================== GPT4 Judgement ====================================
No, the document does not address the question. The document talks about the song "Bounce
with Me" and its connection to the movie Big Momma's House, but it doesn't mention who
plays Big Momma in the movie. The answer to the question is Martin Lawrence, who plays the
character of Big Momma in Big Momma's House.
```

```
==================================== Question ====================================
Who sang "Buddy, Can You Spare a Dime?"
==================================== Answers ====================================
 ['Rudy Vallee', 'Bing Crosby']
==================================== Document ====================================
Brother, Can You Spare a Dime?
Brother, Can You Spare a Dime? "Brother, Can You Spare a Dime?", also sung as "Buddy, Can
You Spare a Dime?", is one of the best-known American songs of the Great Depression.
Written in 1930 by lyricist E. Y. "Yip" Harburg and composer Jay Gorney, "Brother, Can You
Spare a Dime?" was part of the 1932 musical revue "Americana"; the melody is based on a
Russian-Jewish lullaby Gorney's mother had sung to him as a child. It was considered by
Republicans to be anti-capitalist propaganda, and almost dropped from the show; attempts
were made to ban it from the radio.
==================================== Generation ====================================
The song "Brother, Can You Spare a Dime?" was written by E. Y. "Yip" Harburg and Jay Gorney
. It was first performed by Bing Crosby in 1932.
==================================== GPT4 Judgement ====================================
The document provides information about the song "Brother, Can You Spare a Dime?" (also
referred to as "Buddy, Can You Spare a Dime?") and its origins, but it does not directly
answer the question of who sang it. The document focuses more on the history of the song
and its cultural significance.

To answer your question: The song has been performed by several artists, including Bing
Crosby, Al Jolson, and Barbra Streisand, among others.
```

Figure 8: Example of RAG output correct answers while the retrieved document does not contain the correct answer, nor does GPT-4 consider the document address the question.

```
==================================== Question ====================================
When was The Hitchhiker's Guide to the Galaxy made?
==================================== Answers ====================================
['1978', '2005']
==================================== Document ====================================
The Hitchhiker's Guide to the Galaxy (radio series)
The original series was the recipient of a number of awards including the Imperial Tobacco
Award (1978), The Sony Award (1979), The Society of Authors/Pye Awards Best Programme for
Young People (1980) and the Mark Time Awards 'Grand Master Award' (Adams) and 'Hall of Fame
' (1998). It was the only radio show ever to be nominated for the Hugo science fiction
awards, in 1979, in the 'Best Dramatic Presentation' category. As a result of the series,
Douglas Adams was inducted into the Radio Academy's Hall of Fame. In November 2003, two
years after Adams's death and 23 years after the ...

==================================== Generation ====================================
[1] The Hitchhiker's Guide to the Galaxy (radio series)
==================================== GPT4 Judgement ====================================
No, the document does not directly address the query of when The Hitchhiker's Guide to the
Galaxy was made. It provides information about the radio series, its awards, and
recognition but does not specify the year of its creation or first broadcast.

The original radio series of The Hitchhiker's Guide to the Galaxy first aired in 1978.
```

```
==================================== Question ====================================
Who ordered the Forbidden City to be built?
==================================== Answers ====================================
['Zhu Di']
==================================== Document ====================================
History of the Forbidden City
The site of the Forbidden City was situated on the Imperial city during the Mongol Yuan
Dynasty. After the collapse of the Yuan Dynasty, the Hongwu Emperor of the Ming Dynasty
moved the capital from Beijing in the north to Nanjing in the south, and in 1369 ordered
that the Yuan palaces be razed. His son Zhu Di was created Prince of Yan with his seat in
Beijing. In 1402, Zhu Di usurped the throne and became the Yongle Emperor. He made Beijing
a secondary capital of the Ming empire ...

==================================== Generation ====================================
The Yongle Emperor ordered the forbidden city to be built.
==================================== GPT4 Judgement ====================================
Yes, the document addresses the query. It mentions that Zhu Di, who later became the Yongle
 Emperor, ordered the construction of the Forbidden City. The passage describes how he
moved the capital to Beijing and commissioned the building of the Forbidden City in 1406.
```

Figure 9: Example of RAG failing to output the correct answer while the retrieved document contains the correct answer or GPT-4 considers the document as addressing the question.

```
==================================== Question ====================================
Who is the sister of for king and country?
==================================== Answers ====================================
['Rebecca St. James']
==================================== Generation 1 ====================================
Rebecca St. James

### Explanation:
Rebecca St. James is the sister of for king and country. She is a Christian recording
artist and speaker. The Smallbones are the younger brothers of Rebecca St. James, and the
brothers-in-law of Jacob Fink, former bassist for the band Foster the People. They were
raised with another sister, Libby, and three other brothers, Ben, Dan, and Josh. On
==================================== Output 1 ====================================
Eval(Generation): 1
==================================== Generation 2 ====================================
Courtney Helm

### Explanation:
Courtney Helm is the sister of for king and country.
==================================== Output 2 ====================================
Eval(Generation): 0
```

Figure 10: Examples of generations and their evaluation results for free-form tasks.

```
==================================== Question ====================================
Study: Vaccine for Breast, Ovarian Cancer Has Potential.
==================================== Answers ====================================
True
==================================== Generation 1 ====================================
True

### Explanation:
The study is about a vaccine for breast and ovarian cancer. The vaccine is
==================================== Output 1 ====================================
Eval(Generation): 1
==================================== Generation 2 ====================================
False

### Explanation:
The statement is false because the vaccine is not for breast and ovarian cancer.
==================================== Output 2 ====================================
Eval(Generation): 0
```

Figure 11: Examples of generations and their evaluation results for close-set tasks.

```
==================================== Prompt (OpenQA) ====================================
### Paragraph:
[1] For King & Country (band)
Their father was a music promoter, and Joel recalls, "going to these rock concerts, sitting
 on my father\'s shoulders, plugging my ears...Honestly, in a lot of ways I feel like music
 chose me and as I grew older, I made a clear decision to fully lean into it." The
Smallbones are the younger brothers of Christian recording artist and speaker Rebecca St.
James, and the brothers-in-law of Jacob Fink, former bassist for the band Foster the People
. They were raised with another sister, Libby, and three other brothers, Ben, Dan, and Josh
. On ...

### Instruction:
Who is the sister of for king and country?

### Response:
==================================== Continuation ====================================
Rebecca St. James
==================================== Generation ====================================
Rebecca St. James

### Explanation:
Rebecca St. James is the sister of for king and country. She is a Christian recording
artist and speaker. The Smallbones are the younger brothers of Rebecca St. James, and the
brothers-in-law of Jacob Fink, former bassist for the band Foster the People. They were
raised with another sister, Libby, and three other brothers, Ben, Dan, and Josh. On
==================================== Output ====================================
Probility: P('Rebecca St. James'|prompt) = 0.595
Eval(Generation): 1
```

Figure 12: Example prompt and outcomes of each step for NQ and TQA dataset.

```
=============================== Prompt (Fact Checking) =================================
Below is an instruction that describes a task. Write a response that appropriately
completes the request.

### Paragraph:
[1] Gustav Gaudernack
potential of dendritic cells (DCs) and in 2005, Gaudernack's group published results from a
 phase I/II clinical trial in prostate cancer patients using autologous DCs loaded with
tumor mRNA as a vaccine. This study demonstrated that vaccination with autologous DCs
transfected with mRNA derived from three prostate cancer cell lines was safe and an
improved clinical outcome was significantly related to immune responses against the vaccine
. Furthermore, Gaudernack and colleagues initiated a phase I/II clinical trial for
treatment of malignant melanoma with autologous tumor-mRNA transfected DC vaccines. These
data clearly demonstrated vaccine-specific immune responses with a broad specter of ...

### Instruction:
Is the following statement correct or not? Say true if it's correct; otherwise say false.

### Input:
Study: Vaccine for Breast, Ovarian Cancer Has Potential

### Response:
==================================== Continuation ======================================
True
==================================== Generation ========================================
true

### Explanation:
The study is about a vaccine for breast and ovarian cancer. The study has ...
===================================== Output ===========================================
P('true' |prompt) = 0.116
P('false'|prompt) = 0.109
Eval(Generation): 1
```

Figure 13: Example prompt and outcomes of each step for the Pubhealth dataset.

```
=========================== Prompt (Multi-choice Reasoning) ===========================
Below is an instruction that describes a task. Write a response that appropriately
completes the request.

### Paragraph:
[1] Rheumatic fever
Rheumatic fever may occur following an infection of the throat by the bacterium "
Streptococcus pyogenes". If the infection is untreated rheumatic fever can occur in up to
three percent of people. The underlying mechanism is believed to involve the production of
antibodies against a person\'s own tissues. Due to their genetics, some people are more
likely to get the disease when exposed to the bacteria than others. Other risk factors
include malnutrition and poverty. Diagnosis of RF is often based on the presence of signs
and symptoms in combination with evidence of a recent streptococcal infection. Treating
people who have strep ...

### Instruction:
Given four answer candidates, A, B, C and D, choose the best answer choice.

### Input:
Which factor will most likely cause a person to develop a fever?
A: a leg muscle relaxing after exercise
B: a bacterial population in the bloodstream
C: several viral particles on the skin
D: carbohydrates being digested in the stomach

### Response:
==================================== Continuation ======================================
B
==================================== Generation ======================================
B

### Explanation:
The bacteria Streptococcus pyogenes is a common cause of throat
==================================== Output ======================================
P('A'|prompt) = 0.121
P('B'|prompt) = 0.309
P('C'|prompt) = 0.061
P('D'|prompt) = 0.100
Eval(Generation): 1
```

Figure 14: Example prompt and outcomes of each step for the ARC-Challenge dataset.

