# OpenReview forum: "Optimizing Retrieval for RAG via Reinforcement Learning"
_NeurIPS.cc/2025/Conference — NeurIPS 2025 poster_

### Official Review · Reviewer_3W4Q · 2025-07-01

**Clarity:** 3
**Significance:** 3
**Originality:** 2
**Rating:** 3
**Confidence:** 3

**Summary:**

This paper proposes a framework called Open-RAG, which constructs contrastive learning supervision for retriever training by automatically determining whether a given Query-Document pair is a positive match in the context of RAG. This allows the retriever to better align with the downstream needs of the LLM-based QA task, rather than relying solely on semantic similarity, thereby improving task-specific RAG performance.

**Questions:**

1 In Section 3.2, how is the evaluation conducted to construct the positive pool Dq+ and the negative pool Dq−? I guess it might be based on an exact match, but this should be clearly specified in the main text.

2 As a core design choice of this work, the loss terms described in Section 3.4 are critical. However, I find their respective roles insufficiently explained. A more intuitive explanation would be helpful, especially regarding this confusing point: the component EBoT(q) does not participate in inference and appears only in the loss term. What is the rationale for including it in training?

3 In Appendix D, what are the specific procedures for the IR setting and RAG setting respectively? The descriptions of “Unsupervised Pre-training,” “Supervised on MS MARCO,” “Supervised on NQ,” “Supervised on TriviaQA,” and “Pre-training + Supervised on Multiple Datasets” are unclear. Please clarify what each of these means in practice.

4 In Section 4.1, the statement “Our experiments are conducted with 4 NVIDIA GPUs” should include the specific GPU model for reproducibility and transparency.

**Ethical Concerns:**

["NO or VERY MINOR ethics concerns only"]

**Final Justification:**

I would like to maintain my current score after reviewing the response and the feedback from other reviewers.

**Limitations:**

See Questions.

**Paper Formatting Concerns:**

No major formatting issues were found in this paper.

**Quality:**

3

**Strengths And Weaknesses:**

**Strengths:**

1 The paper presents an end-to-end mechanism to optimize the RAG system in accordance with the requirements of the downstream task. Specifically, it uses generation certainty as a signal to construct contrastive supervision for training the retriever. This is an intuitive, effective, and elegant approach.

2 As an online, lightweight, and end-to-end optimization framework for RAG, the design of Open-RAG offers valuable engineering insights.

3 Open-RAG only requires a one-time offline index construction over the entire corpus, significantly reducing system overhead.

**Weaknesses:**

1 The most novel aspect of this paper—its semi-parametric design—has already been proposed in prior work (SIDR), which diminishes the originality of the contribution.

2 The methodological innovation appears limited. Steps 1 and 3 appear to be directly borrowed from SIDR, leaving only Step 2 as the novel component. As a result, the paper feels more like an application of SIDR to RAG, rather than a substantially new contribution.

---

> ### Author Rebuttal · Authors · 2025-07-30
>
> We thank Reviewer 3W4Q for their detailed and insightful review! We would like to address their concerns below:
>
> ---
> > **W1**: The most novel aspect of this paper—its semi-parametric design—has already been proposed in prior work (SIDR), which diminishes the originality of the contribution.
>
> > **W2**: The methodological innovation appears limited. Steps 1 and 3 appear to be directly borrowed from SIDR, leaving only Step 2 as the novel component.
>
> **A1**: We understand the reviewer's concern regarding the originality of our method due to the direct application of SiDR. However, we would like to clarify the following points to demonstrate the contribution of our work.
>
> **Main Contribution Difference**: The primary contribution of our work lies in the optimization of the retriever specifically for RAG workflows. While we leverage SiDR to address the index refreshment issue, this is only one component of improving RAG workflows. There are several other innovations and contributions beyond this. In contrast, SiDR's main contribution is centered on indexing efficiency and cost, with limited empirical experiments focused on RAG.
>
> **Contibutions beyond SiDR**: Please allows us highlight other key contributions.
> -  Section 1 and Appendices B/C/D: We systematically explore and analyze the gap between traditional IR and RAG, identifying current challenges and emphasizing the potential for improving retriever performance in RAG systems.
> - Section 3.3: We propose a threshold-based approach to reduce online LLM autoregressive generation costs, cutting the 100 LLM forward passes (with each generation having a maximum of 100 tokens) down to a single forward pass, making the approach transition from unaffordable to practical.
> - Section 3 and Appendices E/F: We explore different sampling strategies and propose cache implementations to further reduce computation.
> - Section 4: We validate the effectiveness of our overall RAG system and assess its transferability to other LLM systems.
> - Section 5: We conduct ablation studies on different co-training strategies and index refreshment methods.
>
> All these contributions are original to our work, and we believe they will be valuable for future research in this area. While SiDR forms part of our methodology, we hope that the broader scope of our contributions will be recognized, as they go beyond this adoption.
>
> ---
>
> > **Q1**: In Section 3.2, how is the evaluation conducted to construct the positive pool Dq+ and the negative pool Dq−? I guess it might be based on an exact match, but this should be clearly specified in the main text.
>
> **A2**: To clarify, Section 3.2 discusses the retrieval process but does not specifically address how the positive and negative pools are constructed. The details of how we determine the positive and negative documents are provided in Section 3.3. Specifically:
> - For offline positive and negative pools, we evaluate whether the RAG generation contains the answer string (see `L193-195`).
> - For online positive and negative pools, we use a threshold-based approach, which is introduced in Section 3.3 (see `L196-203`).
> - Our evaluation follows prior RAG research (e.g., Self-RAG), where accuracy is used as the metric, based on whether the RAG generation contains the correct answer (see `L236-238`).
>
> We also provide an example in **A1** of our response to `Reviewer rPbd`, which demonstrates the entire calculation process. We kindly recommend reviewing this for further clarification.
>
>
>
> ---
>
> > **Q2**: A more intuitive explanation would be helpful, especially regarding this confusing point: the component EBoT(q) does not participate in inference and appears only in the loss term. What is the rationale for including it in training?
>
> **A3**: Thank you for the insightful question! You're correct that the second term $L(E_{BoT}(q), E_{θ}(d))$ does not appear during inference. Here's a breakdown of the rationale and empirical support for including this term in training:
>
> - **Rationale**: The second term would encourage $E_θ(d)$ to activate lexical tokens from $BoT(q)$. By doing so, it enhances the ability of $E_θ(q)$ to search over $E_θ(d)$, as $E_θ(q)$ typically assigns high term weights to lexical tokens from $BoT(q)$.
>
> - **Empirical support**: In SiDR's rebuttal on ICLR, they empirically demonstrated that this term improves retrieval accuracy. Below are part of their ablation results:
>
> | Model  | NQ   |  TQA   | WebQ |
> |---------------|------|-------|------|
> | SiDR     | 49.1 | 56.2  | 40.2 |
> | SiDR (w/o second term)  | 48.7 | 56.0  | 38.7 |
> | SiDR$_{\beta}$      | 39.8 | 50.4  | 32.1 |
> | SiDR$_{\beta}$  (w/o second term)  | 37.1 | 46.8  | 30.4 |
>
> Additionally, we conducted ablation studies using our OpenRAG method with 1-shot on NQ and PubHealth, where we observed a similar improvement, with the inclusion of the second term providing a noticeable benefit:
>
> | Model  | NQ  |  PubHealth |
> |---------------|------|-------|
> | OpenRAG     | 47.8 | 69.5  |
> | OpenRAG (w/o second term)  | 45.1 | 63.2  |
>
> ---
>
> > **Q3**: In Appendix D, what are the specific procedures for the IR setting and RAG setting respectively? The descriptions of “Unsupervised Pre-training,” “Supervised on MS MARCO,” “Supervised on NQ,” “Supervised on TriviaQA,” and “Pre-training + Supervised on Multiple Datasets” are unclear. Please clarify what each of these means in practice.
>
> **A4**: We apologize for any inconvenience caused. As noted in Appendix D (`L696-698`), the details you refer to are already outlined in Section 4.1 and Appendix C. While this information is mentioned, we understand that it might be scattered. To better address your concern, we will consolidate and repeat this information in Appendix D for the convenience of future readers.
>
> Here are the relevant details with references from the paper:
>
> - For Evaluation metric (details in Section 4.1, `L236-238`):
>     - **IR setting** assesses whether the retrieved document contains the answer.
>     - **RAG setting** assesses whether the RAG generated response contains the answer.
>
> - For Retrieval Model Details (details in Appendix C1, `L638-653`):
>     - **Unsupervised Pre-training** refers to a retrieval model that is pre-trained on unsupervised data without any access to any downstream supervised training data (i.e., human-labeled query-document pairs).
>     - **Supervised on MS MARCO / NQ / TriviaQA** refers to models are fine-tuned on the training data of the MS MARCO, NQ, or TriviaQA datasets, respectively.
>     - **Pre-training + Supervised on Multiple Datasets** combines unsupervised pre-training followed by supervised fine-tuning on multiple datasets. Typically, this combination achieves state-of-the-art performance but comes with significant costs.
>
> ---
>
> > **Q4**: In Section 4.1, the statement “Our experiments are conducted with 4 NVIDIA GPUs” should include the specific GPU model for reproducibility and transparency.
>
> **A5**: We use 4 NVIDIA A100 GPUs, each with 80GB memory. We will include this detail in revision.
>
> ---
>
> We truly appreciate your comments and suggestions and will incorporate them into our final draft. We hope our response has addressed your concerns. Please feel free to let us know if you have any further questions.

---

> > ### Comment · Reviewer_3W4Q · 2025-08-04
> >
> > Thanks for the authors' clarification. Except for A1, the responses to my other questions have addressed my concerns.
> >
> > **A1 (response to authors' A1):**
> > The authors present five points of "Contribution beyond SIDR," among which:
> >
> > *Section 3.3: We propose a threshold-based approach to reduce online LLM autoregressive generation costs, cutting the 100 LLM forward passes (with each generation having a maximum of 100 tokens) down to a single forward pass, making the approach transition from unaffordable to practical.*
> >
> > Apart from this specific point, I believe the remaining parts are essential for demonstrating the effectiveness of the proposed method, although they don't introduce significant theoretical innovation. I have no objections to these parts, and I am satisfied with the contribution to the paper's narrative.
> >
> > However, regarding this particular claimed contribution, the strategy of using a threshold to reduce multiple LLM calls is not new in RAG or reranking systems. Previous work has already adopted thresholding or early stopping to reduce reranking or generation overhead [R1]. Therefore, this approach appears to be more of a common optimization rather than a unique invention.
> >
> > My sole concern is that this work is built upon SIDR, and from my perspective, the methodological novelty is limited—primarily in that SIDR focuses on text matching relevance, whereas OPEN-RAG focuses on end-to-end RAG effectiveness. The main additional solutions that OPEN-RAG introduces on top of SIDR are:
> > - Replacing the retriever’s training objective (from text matching relevance → generation-effectiveness-driven).
> > - Integrating the retriever into the RAG pipeline and adjusting prompts/ranking to ensure the retrieved results are better aligned with LLM generation.
> > I consider these two approaches to be relatively straightforward and intuitive engineering innovations. While I appreciate the engineering value of implementing them in practice, I would expect to see deeper methodological insights. For example, when shifting the goal from building a semi-parametric retriever to applying it in RAG scenarios, are there fundamentally new theoretical challenges? Does this shift require any non-trivial or counterintuitive measures?
> >
> > Once again, I thank the authors for their careful and detailed responses. Regrettably, it remains difficult for me to justify a raised score.
> >
> > [R1] Meng C, Arabzadeh N, Askari A, et al. Ranked list truncation for large language model-based re-ranking[C]//Proceedings of the 47th international ACM SIGIR conference on research and development in information retrieval. 2024: 141-151.

---

> > > ### Author Response · Authors · 2025-08-06
> > > **Response to Reviewer 3W4Q (1/2)**
> > >
> > > We greatly appreciate Reviewer 3W4Q’s detailed and timely feedback, which has provided further insights and valuable directions for discussion. In this response, we will direct response to reviewer's concerns.
> > >
> > > ---
> > >
> > > ### Regarding Initial Rebuttal
> > >
> > > > Thanks for the authors' clarification. Except for A1, the responses to my other questions have addressed my concerns.
> > >
> > > We are glad to have addressed most of the reviewer’s concerns.
> > >
> > > We understand that the main concern regarding A1 and the semi-parametric retriever may stem from the limited technical innovation of these components, as they are not particularly groundbreaking. We would like to provide a broader perspective to justify.
> > >
> > > ---
> > >
> > > ### Regarding SiDR Component
> > >
> > > We appreciate the reviewer’s thoughtful attention to the semi-parametric retriever’s role in the RAG setting.
> > >
> > > While these components are chosen for practical effectiveness rather than novelty, we would like to share the **insights and challenges** we encountered when directly deploying SiDR in RAG training, in order to provide a more comprehensive assessment.
> > >
> > > 1. **The inherent lexical matching of sparse retrievers prevents effective in-training retrieval on new distributions within the new environment**.
> > >
> > > Sparse retrievers rely heavily on both the query and document encoder activating the same relevant tokens. For semi-parametric retrievers, superior effectiveness is attributed to query expansion (as noted in the SiDR paper). However, when deployed in a new RAG environment, query expansion may be suboptimal, and gradients are unable to propagate due to sparsity. We find offline warmup helps effective subsequent online learning.
> > >
> > > We further analyzed token overlap between queries and documents after expansion (i.e., the number of co-activated dimensions between $E_{\theta}(q)$ and $E_{\theta}(p)$), with results as follows:
> > >
> > > | Method | Offline Warmup | Co-activation | Accuracy |
> > > |:-|:-:|:-:|:--:|
> > > | i. SiDR-MS     | No  | 14.3%   |  34.4  |
> > > | ii. OpenRAG (SiDR-MS) | No  | 16.5%   |  37.4 |
> > > | iii.  OpenRAG (SiDR-MS) | Yes |  24.5% | 39.8  |
> > >
> > > As shown by these results, **directly deploying SiDR** (method ii) for in-training retrieval leads to suboptimal RAG performance. Co-activation analysis indicates that offline warmup enables the query and document encoders to activate more relevant dimensions, thereby facilitating effective training and smoothly bridging both the domain gap (MS to NQ) and the setup gap (traditional retrieval to RAG).
> > >
> > > 2. **Trivially using the retrieval scores from SiDR is less effective**; the key is to establish a robust reward function on it.
> > >
> > > Directly using SiDR retrieval and ranking does not lead to significant improvement. We conducted additional experiments to explore this. Suppose there are $k$ documents in the retrieved pool above the positive threshold and classified as positives. OpenRAG (Double) and OpenRAG (Half) take $2k$ and $k/2$ highest-probability documents as positives, resulting in double or half the size of the online positive pool, respectively. The same applies to the negative pool. On the other hand, OpenRAG (KLD) aligns the relevance distribution with the RAG probability distribution.
> > >
> > > | Method | NQ | PubHealth |
> > > |:-|:-:|:-:|
> > > | i. OpenRAG | 39.8  | 69.5   |
> > > | ii. OpenRAG (Double) | 37.2 | 66.8  |
> > > | iii. OpenRAG (Half) | 38.7 | 68.2  |
> > > | iv. OpenRAG (KLD) | 38.2 | 66.4 |
> > >
> > > The results demonstrate that simply aligning relevance with probability (i.e., KLD) is less effective. A threshold-based approach, which incorporates downstream evaluation to determine acceptance and rejection ranges for probabilities,  as a reward function, is more effective. The (Half) and (Double) experiments further show that the chosen threshold range is a well-justified.
> > >
> > > ---
> > >
> > > Next, we would like to share a broader perspective on our work to further justify its novelty and significance within the area.

---

> ### Author Response · Authors · 2025-08-06
> **Response to Reviewer 3W4Q (2/2)**
>
> We would like to provide a broader perspective on our work in this response (2/2).
>
> ---
>
> ## Overview of Our Work
>
> **Overview**: At a high level, our work enables the retriever to interact within the entire RAG environment (LLM, task instructions, and datastore) to explore and self-improve, **aligning the training objective with downstream RAG inference**.
>
> **Challenge**: The major challenge lies in bridging supervision signals from the RAG end results, through the LLM, back to the retriever. This supervision propagation path is both non-differentiable and computationally expensive, due to asynchronous index updates and LLM autoregressive generation. We address these issues using semi-parametric retrieval and a threshold-based solution, respective
>
> ---
>
> ## Search for Users v.s. LLM
>
> - **[A: Learned Searching for User]** Traditional retriever training relies on relevant text pairs labeled by humans or LLMs to verify if the document addresses the query.
>
> However, a document that is relevant to or addresses the query does not necessarily enable the LLM to generate a preferred response. We have conducted preliminary experiments and connected recent findings from different areas to support this point.
>
> - **[B: Learned Searching for LLM]** Our approach leverages the RAG workflow to assess whether a document prompts the LLM to generate a preferred response, aligned with downstream RAG inference.
>
> While methods like RePlug and RA-DIT also follow [B], they primarily optimize for perplexity. In contrast, our method focuses directly on downstream evaluation metrics, which we believe may provide a more direct and end-to-end alignment with task objectives.
>
> ---
>
> ## Highlight
>
> We would like to highlight several key points:
>
> 1. **The novelty of our work lies at the methodological level** -- a shift from [A: Learned Searching for User] to [B: Learned Searching for LLM]. The major challenges are the infeasible training cost and the non-differentiable workflow. We introduce key components specifically designed to address these issues.
>
> We selected these components based on their practical applicability and empirical effectiveness, aiming to address the challenges in real-world application settings. We hope our contribution can be appreciated from a broader methodological perspective, as the overall value may extend beyond the novelty of individual components.
>
> 2. **Regarding contribution**, our work presents a distinct and practical solution for application compared to existing approaches.
>
> There are few existing works focused on optimizing retrievers for RAG, with notable examples including Atlas, RePlug, and RA-DIT. These methods require significant training costs, making them resource-intensive, difficult to generalize to diverse tasks, and challenging to scale to larger datastores. In contrast to resource-intensive methods, our approach is designed to be efficient and lightweight, enabling stable improvements with minimal overhead (e.g., 4 GPUs within a day).
>
> 3. **Regarding novelty**, our work provides a complementary perspective to existing approaches.
>
>     - Baselines that improve retrieval with LLM reasoning differ from our approach, where the retriever remains frozen or is not jointly trained with LLMs.
>     - For baselines that co-train the retriever with LLMs (e.g., RePlug, RA-DIT, Atlas), key technical differences include index refresh strategies, alignment methods, and objectives. Methodologically, our optimization is directly based on RAG outcomes, as opposed to PPL-related objectives in other methods.
>
> ---
>
> ## Drawing an Analogy of Our Work to LLM Post-training
>
> Lastly, we offer a more vivid analogy from the LLM post-training domain to aid understanding:
>
> - Methods such as RePlug, RA-DIT, and Atlas, which use **KLD + PPL objectives**, are analogous to **supervised fine-tuning (SFT)**, as there is an oracle distribution of query-document relevance to align with (i.e., PPL-related distribution from frozen LLMs).
> - In contrast, OpenRAG, which employs **contrastive learning + thresholds**, is more akin to **reinforcement learning (RL)**, where the threshold-based approach serves as an efficient **reward model (RM)** for online training.
>
> OpenRAG operates on-policy, interacting with the entire RAG environment and self-improving based on feedback from this reward model, enabling exploration of new relevance and facilitating alignment with downstream requirements.
>
> Similar to LLM post-training, the key is to effectively and efficiently bridge supervision signals to align with downstream preferences. In practice, as long as the reward model is efficient and reliable, it can serve the purpose well, even without relying on highly complex techniques.
>
> ---
>
> We appreciate the reviewer’s comments regarding technical novelty and hope that this broader perspective helps clarify the significance of our work. We welcome any further suggestions or questions that could help us improve it further.

---

### Official Review · Reviewer_rPbd · 2025-07-03

**Clarity:** 3
**Significance:** 3
**Originality:** 3
**Rating:** 4
**Confidence:** 3

**Summary:**

This paper presents an end-to-end RAG framework that enhances performance by focusing solely on improving the retriever component. Positive and negative query-document pairs for contrastive learning are selected based on the likelihood of generating the final response. The study also highlights that relevance models used in traditional Information Retrieval (IR) often do not align well with RAG performance. Experimental results on five public RAG benchmarks show that the proposed method outperforms other compared approaches.

**Questions:**

Please see my questions in the weakness section. The final score may be changed based on the author response. My main concerns are the setting of thresholds and the ablation study on the objective loss.

**Ethical Concerns:**

["NO or VERY MINOR ethics concerns only"]

**Final Justification:**

Based on the authors rebuttal, which solves my concern in this paper, I keep my original rating.

**Limitations:**

yes

**Quality:**

3

**Strengths And Weaknesses:**

Strengths:
1. The paper introduces an end-to-end optimized RAG framework that enhances RAG performance by tuning the retriever component through online contrastive learning.

2. It demonstrates that higher accuracy in traditional IR settings generally correlates with improved RAG performance for QA tasks, although this correlation does not extend to non-QA tasks.

3. The proposed method outperforms other compared approaches on five public datasets.

Weaknesses:
1. The settings for the thresholds (\tau_q^{+/-}) in line 197 are not clearly explained. These thresholds are crucial for determining positive and negative documents during online contrastive learning. The rationale for using the top 100 documents to establish these parameters is unclear. It would be beneficial to show the impact of varying these parameter settings on performance.

2. An ablation study for the final loss function, as mentioned in line 224, is necessary. It would be insightful to observe the performance differences when different components are included in the objective loss. This analysis would help in understanding the contribution of each component to the overall effectiveness of the model.

---

> ### Author Rebuttal · Authors · 2025-07-30
>
> We thank Reviewer rPbd for their thorough and insightful review! We would like to address their concerns below:
>
> ---
>
> > **W1(1)**: The settings for the thresholds ($\tau_q^{+/-}$) in line 197 are not clearly explained. These thresholds are crucial for determining positive and negative documents during online contrastive learning.
>
> **A1**: Thank you for the feedback! To clarify the intuition, let's use a simple example. Given a query $q$, we offline retrieve $k=5$ documents {$d_1$, $d_2$, $d_3$, $d_4$, $d_5$} to establish the online thresholds. Note that we use $k=100$ in our experiments, and $k=5$ here for illustration.
>
> |        |  \$d_1\$  |  \$d_2\$  |  \$d_3\$  |  \$d_4\$  |  \$d_5\$  |
> |-------|:----:|:----:|:----:|:----:|:----:|
> | \$Gen(q,d)\$   |  1   |  1   |  1   |  0   |  0   |
> | \$P(y \mid q, d)\$  | 0.74 | 0.63 | 0.89 | 0.65 | 0.59 |
>
> In this table, $Gen(q,d)$ indicates whether the RAG generation contains the answer (i.e., pass/fail evaluation), and $P(y \mid q, d)$ refers to the joint probability of ground truth $y$ given $q$ and $d$ (see `L191-193`). Below is the process used to determine positives and negatives online:
>
> - Offline positives = {$d_1$, $d_2$, $d_3$}; Offline negatives = {$d_4$, $d_5$}.
> - Thresholds for online positives $\tau_q^{+}$ = max{0.65, 0.59} = 0.65; Thresholds for online negatives $\tau_q^{-}$ = min{0.74, 0.63, 0.89} = 0.63.
> - In practice, we also use $\tau_q^{+}$ = max{$\tau_q^{+}$, $\tau_q^{-}$} and $\tau_q^{-}$ = min{$\tau_q^{+}$, $\tau_q^{-}$} to ensure $\tau_q^{+}$ > $\tau_q^{-}$.
> - During online training, a document $d_i$ is considered:
>     - **Positive** if $Prob(y \mid q, d_i) > \tau_q^{+}=0.65$. **Rationale**: based on offline $k$ observations, we never see a document $d$ with $Prob(y \mid q, d)$ greater than $\tau_q^{+}=0.65$ that fails the evaluation.
>     - **Negative** if $Prob(y \mid q, d_i) < \tau_q^{-}=0.63$. **Rationale**: based on the offline $k$ observations, we never see a document $d$ with $Prob(y \mid q, d)$ smaller than $\tau_q^{-}=0.63$ that passes the evaluation.
>     - **Discarded** otherwise. **Rationale**: such cases are ambiguous, and classifying them as either positive or negative could risk negatively impacting the training.
>
> ---
>
> > **W1(2)**: The rationale for using the top 100 documents to establish these parameters is unclear. It would be beneficial to show the impact of varying these parameter settings on performance.
>
> **A2**: The choice of $k=100$ is primarily driven by cost-effectiveness. If $k$ is too small, we have insufficient observations to determine the threshold and may need to discard more training queries — see `L196`, where we discard queries that have neither offline positives nor offline negatives to guarantee both pools are non-empty for contrastive learning. On the other hand, if $k$ is too large, it would increase the cost and workload for preparation with limited improvement.
>
> To address your concerns, we conduct further experiments on NQ and PubHealth with varying $k$:
>
> |   | k=5  |  k=20 | k=100 | k=200 |
> |:--:|:--:|:--:|:--:|:--:|
> | NQ | 42.3 | 46.9  | 47.8 | 47.9 |
> | PubHealth | 64.1 | 65.7  | 69.5| 69.7 |
>
> The ablation results support our analysis that increasing $k$ from 100 to 200 provides minimal improvement, while performance drops when $k$ is too small (e.g., 5, 20).
>
> ---
>
> > **W2**: An ablation study for the final loss function, as mentioned in line 224, is necessary. It would be insightful to observe the performance differences when different components are included in the objective loss. This analysis would help in understanding the contribution of each component to the overall effectiveness of the model.
>
> **A3**: The final loss function is derived directly from SiDR, as we adopt its in-training retrieval mechanism. The loss function consists of two components: $L_{para}$ and $L_{semi-para}$.
> - $L_{para}$ is the standard contrastive learning loss that aligns the query embedding with the document embedding.
> - $L_{semi-para}$ is unique to SiDR:
>     - The first term $L(E_{\theta}(q), E_{BoT}(d))$ aligns the embedded query with the tokenized document, which guarantees in-training retrieval with bag-of-token index.
>     - The second term $L(E_{BoT}(q), E_{\theta}(d))$ aligns the tokenized query with the embedded document. Although this term may seem irrelevant, empirical results show that including it provides a beneficial effect.
>
> To address your concern, we present the following ablation results reported in SiDR’s original work, demonstrating the benefit of the second term:
>
> | Model  | NQ   |  TQA   | WebQ |
> |---------------|------|-------|------|
> | SiDR     | 49.1 | 56.2  | 40.2 |
> | SiDR (w/o second term)  | 48.7 | 56.0  | 38.7 |
>
> We also conduct ablation studies using our OpenRAG method on both a QA task (NQ) and a non-QA task (PubHealth). The results consistently show that both terms contribute to performance:
>
> | Model  | NQ  |  PubHealth |
> |---------------|------|-------|
> | OpenRAG     | 47.8 | 69.5  |
> | OpenRAG (w/o second term)  | 45.1 | 63.2  |
>
> ---
>
> > **C1**: Please see my questions in the weakness section. The final score may be changed based on the author response. My main concerns are the setting of thresholds and the ablation study on the objective loss.
>
> **A4**: We appreciate your thoughtful comments. We hope our responses to W1 and W2 have addressed your concerns. Please feel free to let us know if anything remains unclear. We would greatly appreciate it if you could take these clarifications into account when finalizing your score.

---

> > ### Comment · Reviewer_rPbd · 2025-08-07
> >
> > Thank you to the authors for their detailed response, which addressed  my questions.  I keep my original rating.

---

> > > ### Author Response · Authors · 2025-08-07
> > >
> > > We are glad to know that our response addressed your questions. Thank you for your engagement and for taking the time to review our work!

---

### Official Review · Reviewer_shNy · 2025-07-05

**Clarity:** 3
**Significance:** 2
**Originality:** 2
**Rating:** 4
**Confidence:** 3

**Summary:**

The paper studies to apply the previously proposed sparse retrieval model, SiDR, to improving the end-to-end training efficiency of RAG systems.

**Questions:**

1. Why there is only one-shot effectiveness shown in HotpotQA, while all the other datasets show both 1-shot and 10-shot effectiveness?

**Ethical Concerns:**

["NO or VERY MINOR ethics concerns only"]

**Final Justification:**

Thanks for you clarifying and I agree that although the study mainly based on SiDR; the study is still valuable in the community of IR and RAG. I'll increase my score to 4.

**Limitations:**

See Weakness 2 and 3.

**Quality:**

2

**Strengths And Weaknesses:**

Strengths
1. The paper offers a method to improve the training efficiency of RAG by proposing a walk-around method without conducting computationally expensive indexing in RAG training. The proposed method is effective while using the retrieval model, SiDR.
Weaknesses
1. The biggest concern for me is that the comparison is a bit misleading. As I know, all the other models (e.g., RA-DIT and SELF-RAG) are not fine-tuned on the train set of the targeted datasets (e.g., NQ, TriviaQA, HotpotQA). This means the key point I can learn from the experiment is that fine-tuning RAG (either retriever or LLM) on the targeted dataset can help but the generalization capability is not confirmed. Correct me if I’m wrong.
2. The previous study has proposed different RAG training strategies to address the issue of index updating for dense retrievers (See top-100 reranking and Query-side fine-tuning in Table 4 in [1]). Although the work provides another solution based on the previously proposed sparse retriever, SIDR. The motivation and contribution are a bit limited since the paper only builds upon a specific sparse retriever, SIDR.
3. It seems that the obvious improvement is when applying the technique to pre-trained LLMs rather than instruction fine-tuned ones.

[1] Atlas: Few-shot Learning with Retrieval Augmented Language Models; Gautier Izacard, Patrick Lewis, Maria Lomeli, Lucas Hosseini, Fabio Petroni, Tim  Schick, Jane Dwivedi-Yu, Armand Joulin, Sebastian Riedel, Edouard Grave; arXiv:2208.03299

---

> ### Author Rebuttal · Authors · 2025-07-30
>
> We thank Reviewer shNy for their thorough and insightful review. We would like to address their concerns below:
>
> ---
>
> > **W1**: The biggest concern for me is that the comparison is a bit misleading. As I know, all the other models (e.g., RA-DIT and SELF-RAG) are not fine-tuned on the train set of the targeted datasets (e.g., NQ, TriviaQA, HotpotQA). This means the key point I can learn from the experiment is that fine-tuning RAG (either retriever or LLM) on the targeted dataset can help but the generalization capability is not confirmed. Correct me if I’m wrong.
>
> **A1**: We understand the reviewer may be concerned that our improvement comes solely from access to the supervised training data. We would like to clarify the following:
>
> **Clarification on the train set**. To our knowledge, SELF-RAG leverages the KILT dataset [1] for training, which includes NQ, TQA, and HotpotQA. The baseline using the Llama3-8B instruction-tuned model should be considered as having access to these well-known datasets during its instruction-tuning process. While RA-DIT does not have access to these datasets, it has been trained on various Wikipedia-based QA datasets, which likely benefit performance on NQ, TQA, and HotpotQA as well.
>
> **The role of the training data is not the primary factor**.  Access to supervised training data alone is not the dominant factor. The critical aspect is how training is designed to optimize the retriever for RAG. This is supported by the following observations:
> - (i) standard retriever fine-tuning on supervised data results in less effective RAG performance;
> - (ii) there are other existing strategies for optimizing the retriever for RAG, and even with supervised data, these methods underperform compared to ours (details discussed below).
>
> **Supporting evidence and comparisons**:
> - Standard retriever fine-tuning (Table 3) vs. our method that optimizes the retriever for RAG (Table 1), where both access the training data but OpenRAG leads to significant improvements. This comparison shows that having access to the training data is not the key factor; rather, the crucial aspect is how the data is leveraged to optimize the retriever for the entire RAG system.
>
> - For the "improving LLM for RAG" baselines, we recommend comparing our method to SELF-RAG and Llama3-Instruct-8B + SIDR, where OpenRAG consistently outperforms SELF-RAG and matches Llama3-Instruct-8B + SIDR on certain datasets. These baselines are introduced to give the reader a sense of the relative improvement of optimizing the retriever compared to improving the LLM, as noted in L264.
>
> - For the "improving IR for RAG" baselines (e.g., RePLUG, RA-DIT's retriever), we suggest reviewing our ablation study in Section 5.1. The key differences between OpenRAG and these models are in the alignment objective (Online Contrastive Learning vs. KL-Divergence) and index refresh strategies (Late Parametric vs. Periodic Re-indexing). These ablation experiments can be viewed as variants of RePLUG using the same LLM, instructions, and **training data**. The results show that, among the methods optimizing the retriever for RAG, OpenRAG introduces several constructive improvements, proving it to be superior and more cost-effective.
>
> ---
>
> > **W2**: The previous study has proposed different RAG training strategies to address the issue of index updating for dense retrievers (See top-100 reranking and Query-side fine-tuning in Table 4 in [1]). Although the work provides another solution based on the previously proposed sparse retriever, SIDR. The motivation and contribution are a bit limited since the paper only builds upon a specific sparse retriever, SIDR.
>
> **A2**: Thank you for your detailed question and for pointing out the relevant reference. We would like to address your concern from several aspects below:
>
> **Regarding index staleness issues**. The three index solutions mentioned in the Atlas paper each have inherent limitations:
> 1. Full index updates periodically (e.g., RePlug) incurs high costs, and our ablation study in Section 5.1 shows that it is less effective than ours.
> 2. Top-100 reranking requires partial index updates. This is incompatible with tools like FAISS for dense indexes and torch.sparse for sparse indexes, making it impractical for large datastores. As a result, few works use this approach due to its inefficiency.
> 3. Query-side fine-tuning (e.g., Atlas and RA-DIT) with frozen document encoders and static indices can lead to inconsistent or limited improvements, as outlined in their papers.
>
> Our solution, based on a semi-parametric design, improves effectiveness, reduces training costs, and ensures a straightforward training process, making it easier to deploy for RAG applications. We hope the benefits of our approach are considered alongside its potential limitations.
>
>
> **From a broader perspective**: Our work addresses the optimization of the retriever for the entire RAG system, where index staleness is just one of the challenges we tackle. Other key contributions and insight include:
>
> - Preliminary experiments to highlight the challenges faced by retrievers in the transition from traditional IR to RAG setups.
> - A threshold-based approach to reduce the online LLM autoregressive generation costs, cutting the 100 LLM forward passes (with a maximum of 100 tokens per generation) down to a single forward pass, making the training cost affordable.
> - Exploration of different sampling strategies and the proposal of cache implementations.
> - Validation of the overall effectiveness and transferability of our RAG system.
> - Detailed ablation studies on different co-training strategies and index refresh methods.
>
> We believe these contributions and analyses will be valuable for future research in this area. While SiDR forms part of our methodology, we hope the broader scope of our contributions will be recognized, as they extend beyond this adoption.
>
> ---
>
> > **W3**: It seems that the obvious improvement is when applying the technique to pre-trained LLMs rather than instruction fine-tuned ones.
>
> **A3**: We primarily use pre-trained LLMs because it is very likely that the training data from well-known datasets (e.g., NQ, TQA, HotpotQA) are involved during instruction tuning. Using pre-trained LLMs better aligns with real-world application scenarios, where the online user query is not part of the LLM's training data.
>
> As shown in Table 1, despite OpenRAG being optimized for Llama3-8B, transferring to Llama3-8B-instruct (the instruction fine-tuned version) still results in improved RAG performance.
>
> To better address your concern, we also experimented with our method using Llama3-8B-instruct as the LLM, and the results show that our method can also improve performance for instruction-tuned models:
>
> |  | NQ  |  PubHealth |
> |---------------|------|-------|
> | Llama3-8B-instruct + SiDR     | 47.2 | 67.2  |
> | OpenRAG (Llama3-8B-instruct)  | 49.3 (+2.1) | 69.8 (+2.6) |
>
>
>
> ---
>
> > **Q1**: Why there is only one-shot effectiveness shown in HotpotQA, while all the other datasets show both 1-shot and 10-shot effectiveness?
>
> **A4**: The difference arises from the use of different Wikipedia datastores (`L243-245`). Specifically, HotpotQA uses a Wikipedia corpus containing 5 million documents, with each document being the first paragraph of a Wikipedia page, ranging from 20 to over 1k words. In contrast, all other datasets use the same Wikipedia corpus, which contains 21 million passages, each 100 words long. Using 10-shot for HotpotQA results in contexts that are too long and exceed the maximum context window of the LLM.
>
> ---
>
> Thank you once again for your thoughtful review. We hope our response addresses your concerns and questions. If you have any further inquiries, please feel free to follow up.
>
> ---
>
> **Reference**
>
> [1] Petroni, Fabio, et al. "KILT: a benchmark for knowledge intensive language tasks." ACL 2023.

---

> > ### Comment · Reviewer_shNy · 2025-08-07
> >
> > Thanks for you clarifying and I agree that although the study mainly based on SiDR; the study is still valuable in the community of IR and RAG. I'll increase my score to 4.

---

> > > ### Author Response · Authors · 2025-08-08
> > >
> > > We sincerely appreciate your thoughtful feedback and for acknowledging the value of our work in the IR and RAG community. Thank you also for your updated evaluation.

---

### Official Review · Reviewer_rFXv · 2025-07-20

**Clarity:** 3
**Significance:** 3
**Originality:** 3
**Rating:** 6
**Confidence:** 4

**Summary:**

This work highlights the importance of finetuning retrievers for RAG. They propose a method called OPEN-RAG that finetunes the retriever using contrastive pairs that are selected by running examples “end-to-end” through the full RAG pipeline. This approach shows improvement over other similar methods across standard RAG benchmarks such as NQ and HotPotQA. The authors also explore both online and offline forms of contrastive learning.

**Questions:**

Q1: Many practitioners use models like ChatGPT, Claude and Gemini via APIs.How well will this method work? Presumably it can be adapted to API-based RAG systems? The offline computation of probability thresholds is quite clever (starting line 189). Is this something that wouldn’t be possible with APIs?

Q2: In an API scenario, could you use an LLM to judge/rank relevance to pick positive and negative pairs instead of calculating probabilities? While it might be more costly, it might make the method more generalizable

Q3: Does it matter if the training objective follows SiDR with the semi-parametric contrastive loss? Did you consider any other approaches (I guess KL divergence)?

Other comments:
* The name is very confusing, and does not convey any information about what the method actually does. OPEN-RAG sounds like it is RAG on an open set of questions. The method is also not end-to-end, since it is only optimizing/training the retriever. I really think the authors should consider changing the name to something that conveys a little more information. A good name goes a long way! And a bad name can cause a lot of unnecessary confusion. Even something as simple as “retriever-optimized RAG”
* Section 1 line 89 states that OPEN-RAG offers “two forms of openness" - both of these are silly and frankly distracting (it almost sounds like it was AI generated). I’m sure you can come up with a better name.
* It would be helpful to include the preliminary study mentioned in line 45 in the main text and not in the appendix, if it is indeed very important
* Typo line 243 “defualt”
* Typo line 260 “while frozen the computation”

**Ethical Concerns:**

["NO or VERY MINOR ethics concerns only"]

**Final Justification:**

The authors have adequately addressed my concerns.

**Limitations:**

yes

**Quality:**

3

**Strengths And Weaknesses:**

Strengths:
* This seems like quite a promising method. Finetuning the retriever is also elegant/straightforward and lightweight compared to other methods
* The authors include ablation studies (online vs. offline, KL divergence vs contrastive loss). The appendix has extensive additional details and experiments

Weaknesses:
* It is surprising that offline only and online only alone result in suboptimal performance. The fact that the method relies on a mix of online and offline undermines the elegance of the approach. How much of this just depends on hyperparameters?
* In Table 1, OPEN-RAG shows improvements across 3 different LLMs for NQ, Trivia And HotPotQA but not PubHealth or ARC-C. What is it about these two datasets?
* Data quality usually plays a crucial role in finetuning. Additional experiments around data quality would make the paper stronger (Figures 3 and 4 are helpful)

---

> ### Author Rebuttal · Authors · 2025-07-31
>
> We sincerely thank Reviewer rFXv for their thoughtful and constructive review. We would like to address their concerns below:
>
> ---
>
> > **W1(1)**: It is surprising that offline only and online only alone result in suboptimal performance.
>
> **A1**: Yes, the suboptimal performance from using offline or online training alone is expected. Offline training can be seen as deploying the retriever into a new environment with pre-labeled data, acting as a "warm-up" phase. On the other hand, online training can be considered as a self-improvement process, where the retriever continuously interacts with the environment and refines itself accordingly. The use of a "warmup" phase is typically beneficial, as it helps to stabilize the subsequent online training, leading to better overall performance.
>
> ---
>
> > **W1(2)**: The fact that the method relies on a mix of online and offline undermines the elegance of the approach. How much of this just depends on hyperparameters?
>
> **A2**: We do not focus extensively on hyperparameters related to the online-offline mixture ratio. Instead, we directly adopt SiDR's ratio, which is an equal mix of offline and online (50%-50%).
>
> Regarding the **elegance** of the approach, we did explore alternatives that use online training only. For example, during online training, we retrieve the top-$k$ (with $k$=20). If we were to gradually increase $k$ from 0 to 20 throughout the training, we could still achieve similar results even without offline warmup. This is because, when k is small (e.g., $k$=2), the top-$k$ retrieved pool is unlikely to contain negatives, prompting us to fall back on the offline negative pool (see `L208`), thus achieving a result similar to offline warmup.
>
> We ultimately decided to present the work using a mix of online and offline training for the following reasons:
> 1. Offline training does not require in-training retrieval, making it more efficient and robust. It is also more likely to transfer stably to new tasks and datastores.
> 2. We anticipate that certain users or developers may prefer to use the offline-only to boost their RAG system to a satisfactory level, rather than relying on the more complex offline + online to achieve the utmost performance. Therefore, separating them would make it easier for others to follow up our work.
>
> ---
>
> > **W2**: In Table 1, OPEN-RAG shows improvements across 3 different LLMs for NQ, Trivia And HotPotQA but not PubHealth or ARC-C. What is it about these two datasets?
>
> **A3**: Regarding the transferability across different LLMs, we discuss this in `L287-295`. Additionally, it may be related to how these datasets are constructed. For example, the queries in NQ, TQA, and HotpotQA are based on Wikipedia, and the task is designed to work with the knowledge within Wikipedia corpus. In contrast, PubHealth and ARC are not.
>
> - For NQ, TQA, and HotpotQA, the Wikipedia datastore likely contains documents with clear, direct answers. The relationship between queries and answers is so straightforward that even different LLMs can easily extract the same answer from the document, enabling transferability across LLMs.
>
> - However, for PubHealth and ARC, there may not be a document specifically designed to provide a direct answer to the query, and different LLMs may have varying preferences for which documents should be included in the context, making transferability more challenging.
>
> ---
>
> > **Q1**: Many practitioners use models like ChatGPT, Claude and Gemini via APIs.How well will this method work? Presumably it can be adapted to API-based RAG systems? The offline computation of probability thresholds is quite clever (starting line 189). Is this something that wouldn’t be possible with APIs?
>
> **A4**: We appreciate the insightful question and thank you for your compliment on our threshold-based approach! Since ChatGPT, Claude, and Gemini APIs are closed source, the applicability of our method depends on whether these APIs can output prompt logits. If they do, our method could be adapted to work with these APIs. If not, we could optimize the retriever using open-source LLMs (e.g., DeepSeek and Qwen) with similar intelligence and scale. As discussed in **A3** above, if the datastore is well-designed and relevant, the learned retriever is likely to transfer effectively.
>
> We believe that co-training a retriever with these API-based LLMs is a common challenge for any methods aiming to optimize the retriever through interaction with the LLM. In this context, the transferability analysis we provide offers a unique insight for future research, enabling us to handle such scenarios effectively.
>
> ---
>
> > **Q2**: In an API scenario, could you use an LLM to judge/rank relevance to pick positive and negative pairs instead of calculating probabilities? While it might be more costly, it might make the method more generalizable
>
> **A5**: It is possible, and we are keen to explore this approach. The only limitation is our limited GPU resources and API token budget. Using an LLM as a judge is similar to a reward model, which has proven effective in training LLMs. This could enable the retriever to self-improve (retrieve → reward → update retriever) and address tasks lacking robust evaluation metrics or require expensive human annotation costs.
>
> In practice, there are additional considerations. First, beyond the cost, the efficiency of using API LLMs for online evaluation could be challenging. A more practical solution could be deploying a comparable open-source LLM (e.g., DeepSeek-671B) as a local service to guarantee efficiency. Second, while LLM-as-a-judge has its advantages, it also has biases and limitations. Combining different evaluation metrics for online tranining would provide a more balanced solution.
>
> ---
>
> > **Q3**: Does it matter if the training objective follows SiDR with the semi-parametric contrastive loss? Did you consider any other approaches (I guess KL divergence)?
>
> **A6**: Thank you for the question! Due to page limits, please refer to our **A3** to `Reviewer rPbd`, where we explain each term of our loss function and the rationale behind it, along with ablation experiments. In Section 5.1 (`L313-326`), we compare contrastive loss with KL-divergence and find that contrastive loss performs better.
>
> We use an example to illustrate why. Given a query $q$ and its top-5 retrieved documents, where $Gen(q,d)$ measure whether using $d$ can pass the evalution and $Rel(q, d)$ is the relevance between $q$ and $d$.
>
> |        |  \$d_1\$  |  \$d_2\$  |  \$d_3\$  |  \$d_4\$  |  \$d_5\$  |
> |-------|:----:|:----:|:----:|:----:|:----:|
> | \$Gen(q,d)\$   |  1   |  1   |  1   |  0   |  0   |
> | \$P(y \mid q, d)\$  | 0.74 | 0.63 | 0.89 | 0.65 | 0.59 |
> | \$Rel(q, d)\$  | 3.7 | 2.9 |  2.4 | 0.9 | 0.02 |
>
> In this case, online contrastive objective will try to maximize $Rel(q, d_1) \rightarrow \infty$ and minimize $Rel(q, d_5) \rightarrow -\infty$. However, KL-divergence would try to reduce $Rel(q, d_1)$ and increase $Rel(q, d_5)$, in this way, ensuring the distribution of $Rel(q, d)$ to become more similar to $P(y \mid q, d)$, which is not the desired outcome.
>
> In our experience, using KL-divergence leads to rapid convergence in the early stages (typically the first few epochs) with limited improvement. In contrast, the online contrastive objective continues to improve performance over a longer training period (see our `Figure 3`).
>
> ---
>
> > **C1**: The name is very confusing, and does not convey any information about what the method actually does. ... I really think the authors should consider changing the name to something that conveys a little more information. A good name goes a long way! And a bad name can cause a lot of unnecessary confusion. Even something as simple as “retriever-optimized RAG”
>
> > **C2**: Section 1 line 89 states that OPEN-RAG offers “two forms of openness" - both of these are silly and frankly distracting (it almost sounds like it was AI generated). I’m sure you can come up with a better name.
>
> **A7**: Thank you for the suggestion and encouragement! We agree that while "open" comes from the initials of "OPtimize ENd-to-end," it may lead to misleading preconceptions. To address this concern, we are considering replacing the title with something like "(Consistently) Optimizing Retrieval for RAG" to better highlight the training-inference consistency. Additionally, Section 1, L89 will be restated to focus on contributions beyond just effectiveness, with the term "openness" removed.
>
> ---
>
> > **C3**: It would be helpful to include the preliminary study mentioned in line 45 in the main text and not in the appendix, if it is indeed very important
>
> **A8**: Thank you for the suggestion. If our paper is accepted with 10-page limitation, we will consider moving this content into the main text to better provide the contextual motivation. We will also correct the typo and ensure everything is checked properly.
>
> ---
>
> Thank you once again for your constructive review. We hope our response satisfactorily addresses all your questions. Should you have any further doubts or concerns, please feel free to follow up.

---

### Decision · Program_Chairs · 2025-09-17

**Decision:**

Accept (poster)

**Comment:**

This paper proposes OPEN-RAG, a framework that tunes a retriever for a retrieval-augmented generation (RAG) system. The method uses online contrastive learning. The learning signal comes from the final RAG generation outcome, which aligns the retriever's relevance with the downstream task. The authors use a semi-parametric retriever to make this end-to-end training efficient and avoid costly index updates. The experiments show that this approach improves performance on several RAG benchmarks. It is more cost-effective than tuning the large language model itself.

Strengths. It addresses the important problem of relevance mismatch between information retrieval and RAG. Reviewers praised its elegance and practicality. The empirical evaluation is comprehensive, with thorough ablation studies that validate the design choices.

Weaknesses. Limited methodological innovation - the framework builds heavily on the existing SIDR retriever architecture. This led one reviewer to argue that the contribution is more of an engineering application than a fundamental advance. Another reviewer initially questioned the fairness of the baseline comparisons.

Rebuttal & discussion period. The authors were very responsive. 3W4Q raised a strong objection regarding the limited novelty. The authors countered by providing new ablation studies showing that a naive application of SIDR fails and that their specific design is crucial for success. shNy had concerns about unfair comparisons. The authors clarified the training data used by the baselines and provided new experiments to strengthen their claims. The rebuttal successfully convinced 3/4 reviewers to recommend acceptance.

Why accept? We put more weight on the paper's practical impact. It provides an effective and efficient solution to a critical problem for anyone building RAG systems. Yes, the novelty of individual components is limited, but the complete framework delivers clear and significant value to the community.